# Inadequate structural constraint on Fab approach rather than paratope elicitation limits HIV-1 MPER vaccine utility

Kemin Tan [1,14], Junjian Chen[2,3,10,14], Yu Kaku[2,3], Yi Wang[2,3,11], Luke Donius[2,3,12], Rafiq Ahmad Khan [2,3], Xiaolong Li [2,3,13], Hannah Richter[4], Michael S. Seaman[4], Thomas Walz[5], Wonmuk Hwang[6,7,8], Ellis L. Reinherz [2,3] ✉ & Mikyung Kim[2,9] ✉

Broadly neutralizing antibodies (bnAbs) against HIV-1 target conserved envelope (Env) epitopes to block viral replication. Here, using structural analyses, we provide evidence to explain why a vaccine targeting the membrane-proximal external region (MPER) of HIV-1 elicits antibodies with human bnAb-like paratopes paradoxically unable to bind HIV-1. Unlike in natural infection, vaccination with MPER/liposomes lacks a necessary structure-based constraint to select for antibodies with an adequate approach angle. Consequently, the resulting Abs cannot physically access the MPER crawlspace on the virion surface. By studying naturally arising Abs, we further reveal that flexibility of the human IgG3 hinge mitigates the epitope inaccessibility and additionally facilitates Env spike protein crosslinking. Our results suggest that generation of IgG3 subtype class-switched B cells is a strategy for anti-MPER bnAb induction. Moreover, the findings illustrate the need to incorporate topological features of the target epitope in immunogen design.

High sequence diversity of circulating HIV-1 isolates within and among infected individuals poses a challenge for host immunity. This diversity also convolutes the design of preventive vaccines that may play a critical role in controlling the HIV epidemic[1]. Given that HIV-1 prophylactic vaccines designed to elicit CD8[+] T cell responses and non-neutralizing antibodies have failed to demonstrate protection in clinical efficacy trials[2,3], it is widely believed that an effective vaccine must be precisely fashioned to elicit broadly neutralizing antibodies (bnAbs) capable of binding to conserved epitopes on the HIV-1 trimeric gp160 envelope (Env) and thereby prevent viral replication[4]. On the virion,

gp160 is a glycoprotein spike consisting of three non-covalently associated gp120 and gp41 subunit protomers. These sparsely arrayed gp160 spikes represent the only target for protective humoral immunity[5].

Among regions harboring conserved epitopes, the membrane-proximal external region (MPER) of the gp41 subunit is a prime quarry. The MPER connects the Env ectodomain to its transmembrane (TM) domain and plays an important role in fusion of HIV-1 to host cells, a process inhibited by anti-MPER bnAbs[6]. On the virion surface, the MPER lies on the viral membrane where it is sterically occluded by the

[1]Structural Biology Center, X-ray Science Division, Advanced Photon Source, Argonne National Laboratory, Lemont, IL, USA. [2]Laboratory of Immunobiology, Department of Medical Oncology, Dana-Farber Cancer Institute, Boston, MA, USA. [3]Department of Medicine, Harvard Medical School, Boston, MA, USA. [4]Center for Virology and Vaccine Research, Beth Israel Deaconess Medical Center, Boston, MA, USA. [5]Laboratory of Molecular Electron Microscopy, The Rockefeller University, New York, NY, USA. [6]Department of Biomedical Engineering, Texas A&M University, College Station, TX, USA. [7]Department of Materials Science & Engineering, Texas A&M University, College Station, TX, USA. [8]Department of Physics & Astronomy, Texas A&M University, College Station, TX, USA. [9]Department of Dermatology, Harvard Medical School, Boston, MA, USA. [10]Present address: Laboratory of Immunology, Department of Immunology and Microbiology, Zhongshan School of Medicine, Sun Yat-sen University, Guangzhou, Guangdong, China. [11]Present address: NeoCura Bio-Medical Technology Co., Ltd., Beijing, China. [12]Present address: AbbVie Bioresearch Center, AbbVie Inc., Worcester, MA, USA. [13]Present address: School of Life Sciences, Division of Life Sciences and Medicine, University of Science and Technology of China, Hefei, Anhui 230027, China. [14]These authors contributed equally: Kemin Tan, Junjian Chen. ✉e-mail: ellis_reinherz@dfci.harvard.edu; mikyung_kim@dfci.harvard.edu

gp160 ectodomain elements from above, residing in a 10-Å crawlspace that greatly limits antibody (Ab) access[7–11]. Recent cryo-electron tomography (cryo-ET) along with cryo-electron microscopy (cryo-EM) analyses revealed that spontaneous tilting of the spike[10,11] and conformational motions of the membrane-associated tripod MPER structure provide access to the MPER protomer opposite the tilt direction[10]. MPER binding is further enhanced after gp120 engagement of the CD4 receptor. Although natural anti-MPER immune responses in people living with HIV-1 have been reported to be low compared with other vulnerable antigenic sites in Env, MPER-specific bnAbs can arise after several years of infection in a subset of individuals[12]. These bnAbs manifest outstanding neutralization breadth predicated on recognition of contiguous structural epitopes, particularly those such as 10E8 and DH511 directed at the MPER-C helix[13–19], the distal MPER segment connected to the proximal MPER-N helix via a hinge. The 10E8 and the most potent DH511 clonal lineage bnAb (DH511.2) neutralized 203 of 208 and 206 of 208 viruses (98% and 99%), respectively, in a panel of geographically and genetically diverse HIV-1 Env pseudoviruses with neutralization potency of median $IC_{50}$ at 0.4 µg/ml for 10E8 and 1.0 µg/ml for DH511.2[13,14]. Notably, many bnAbs are of the IgG3 subclass and exploit hydrophobic residues at the apex of their long heavy chain complementarity-determining region 3 ($CDR3_H$) loop to interact with the MPER and vicinal lipids to achieve neutralization breadth and potency[15,20–25]. However, the basis for the prevalence of IgG3 subclass Abs among anti-MPER bnAbs is uncertain given the equivalent anti-viral activity of mature IgG1 and IgG3 subtypes with identical ligand specificity[16,26,27].

Abs recognizing the key linear epitopes in the MPER identified by bnAbs are also elicited by peptide or protein scaffold vaccines and manifest significant binding affinity for the MPER segments. Despite their similar binding characteristics, no neutralizing activity was observed from the vaccine-elicited Abs[28,29]. Host tolerance mechanisms linked to anti-MPER bnAbs reactivity with self-antigens have been suggested as a barrier to the elicitation of bnAbs by vaccination[30]. Here, we demonstrate through structural analyses that the approach angles of the vaccine-elicited MPER Abs are poorly suited to access the MPER crawlspace on the virion surface. Unlike the steric occlusion provided by gp160, the MPER/liposome and protein scaffold vaccines lack a structure-based selection mechanism to promote MPER-specific B cells with requisite approach angles, eliciting instead dominantly non-neutralizing sera Abs unable to bind the MPER of gp160 embedded on a membrane surface. Additional flexibility of the human IgG3 hinge is revealed to be critical for germline-like Abs of this IgG subclass through facilitation of inter-spike crosslinking, thus mitigating the low binding affinity and granting access to the MPER at the same time. These features account for the high prevalence of the numerically minor IgG3 subtype among MPER-specific bnAbs arising in HIV-1 infected individuals. Overall, our observations emphasize the importance of the integration of the linear target epitope with the structural crawlspace topology as essential immunogen components. Our structural findings along with the avidity of those IgG3 subtype class-switched B cells appearing during early host immune response suggest a future strategy with broad implications for vaccine design against infectious pathogens.

## Results

### Characteristics of MPER/liposome vaccine-elicited antibodies

Given the characteristics of MPER-specific bnAbs noted above, a liposome vaccine displaying N-terminally palmitoylated MPER and the contiguous TM domain synthetic peptide (pMPERTM) (Fig. 1a) was tested for its immunogenicity in BALB/c mice. High-titer MPER-specific polyclonal Abs ($EC_{50}$ $\log_{10}$ titer: 4.2) were elicited as assessed by ELISA

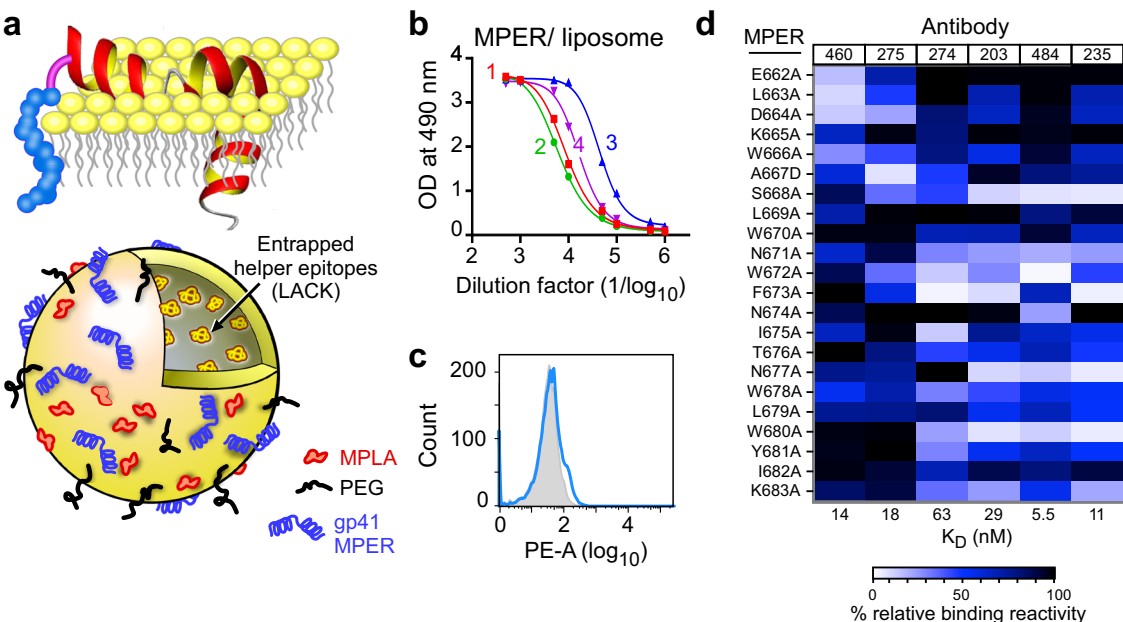

**Fig. 1 | Immunogenicity of MPER/liposome vaccine. a** Schematics of the N-terminally palmitoylated MPER preceding the transmembrane domain of the HIV-1 HxB2 gp41 (pMPERTM) immunogen arrayed on the membrane surface and a standard liposome vaccine formulation including MPLA adjuvant, LACK CD4 T cell helper epitope, PEG-2000 and MPER peptide. The MPER is shown as 2 helices separated by a hinge as was defined by NMR studies[73]. **b** Serum IgG responses of four representative BALB/c mice administered intradermally with 50 µl of pMPERTM/liposome vaccine three times at 3-week interval as described in Methods. The immune sera were collected 30 days after the final injection and MPER-specific IgG responses were determined against Npalm-MPER/liposome by ELISA. **c** gp160 binding of purified polyclonal antibody from combined immune sera of the 4 mice shown in (**b**). 293T cells expressing ADA gp145 were stained with the purified polyclonal antibody at 50 µg/ml (blue line). Histogram shown as filled gray area indicates untransfected negative control cells stained with the same polyclonal antibody. **d** A heatmap comparing epitope specificities of 6 vaccine-elicited rmAbs by Biacore 3000 using liposome-bound single alanine MPER mutants. The horizontal color bar indicates the percent rmAb-binding activity against each single-residue mutant compared to that of wild-type MPER (WT) (100%). Ab name and affinity as determined by Biacore are given in top and bottom lines, respectively. Source data for (**b**) and (**d**) are provided as a Source Data file.

(Fig. 1b) and yet, the vast majority of these Abs, unlike patient-derived bnAbs, was unable to stain 293T cells expressing ADA gp145 trimers (a cleavable ADA wild-type protein truncated at cytoplasmic domain residue A723) (Fig. 1c). To understand the discrepant anti-MPER Ab reactivity, we characterized target epitopes by alanine-scanning mutagenesis to discern if they were distinct from those of previously reported bnAbs. For this purpose, we sorted individual MPER-specific, long-lived single plasma cells from mouse bone marrow 100 days after a final booster immunization, and then performed PCR cloning and produced mouse-human hybrid IgG1 recombinant monoclonal Abs (rmAbs), as previously described[31,32]. Epitopes for six representative rmAbs were mapped using the MPER peptide variants arrayed on liposomes (Fig. 1d). The rmAbs 460 and 275 map to the 2F5 bnAb binding site whereas the remaining four (274, 203, 484 and 235) map to the 10E8/4E10 bnAb epitope cluster. Two rmAbs, 235 and 460, were further selected for structural studies using X-ray crystallography. The former recognizes a MPER-C helix epitope while the latter recognizes an epitope in the MPER-N helix.

## Fab235 approach to MPER-C distinct from bnAbs 10E8 and 4E10

Crystal structures of the Fab fragment of rmAb235 and its complex with the MPER [MPER-C, amino acids (a.a.) $A_{667}$ to $K_{683}$] were determined to resolution limits of 1.94 Å and 2.45 Å, respectively (Supplementary Table 1). There is one Fab235 per asymmetric unit of the *Apo* (i.e., Fab only) crystal but four Fab235/MPER complexes in each asymmetric unit of the *Holo* Fab235 crystal [designated LHP (light chain, heavy chain and peptide), ABC, DEF and GIJ]. In the following description of the Fab235/MPER-C complex, the representative complex LHP is used unless otherwise specified.

CDRs of Fab235 reveal two protruding loops, the heavy chain ($V_H$) FG-loop (CDR3$_H$) and the long BC loop (CDR1$_L$) of the kappa light chain ($V_L$) (Fig. 2a). CDR3$_H$ (a.a. A97 to V112) is rich in tyrosine residues including a variant binary YS sequence motif known to facilitate Ab binding ($Y_{101}$YYGSSYYY$_{109}$) (PDB: 1ZA3)[33]. The Fab235$V_H$ participates dominantly in binding to the helical portion of MPER-C that runs through a shallow groove between CDR3$_H$ and the C′C″-loop (CDR2$_H$). This interface is distinct from those of 4E10 and 10E8 (Supplementary Figs. 1, 2 and Supplementary Table 2). The Tyr- and Ser-rich $V_H$ FG-loop forms the major part of the paratope (Fig. 2b). There are two hydrogen bonds between MPER-C and the FG-loop (N674$_{MPER}$ to Y107$_H$ and N677$_{MPER}$ to Y101$_H$). Water-mediated interactions between Fab235 and the MPER include linking N674 to S106$_H$ and G104$_H$ (Supplementary Fig. 3). Also, the sidechain of Y681$_{MPER}$ packs onto the flat side of a type I′ β-turn formed by $Y_{102}$YGS$_{105}$ with the stacking of aromatic sidechains onto the mainchain involving amide-π and CH-π interactions[34]. The indole ring of W680$_{MPER}$ is similarly stacked on a relatively flat side of a small helical motif from G54$_H$ to S55$_H$ of CDR2$_H$. Additionally, the sidechain of N677$_{MPER}$ forms a hydrogen bond to N52$_H$ on CDR2$_H$ and to a conserved water molecule found in all four complexes (Supplementary Fig. 3). T676$_{MPER}$ does not interact with Fab235 directly, but indirectly through a hydrogen-bond network. A part of the C″-strand of $V_H$ also participates in the interaction, including the I57$_H$ to F673$_{MPER}$ hydrophobic contact (Fig. 2b). In contrast to the Fab235$V_H$ and despite its protruding CDR1$_L$, Fab235$V_L$ makes no direct contact to the MPER aside from a single hydrogen bond (N99$_L$ to S668$_{MPER}$) observed in two of the four *Holo* conformations (Supplementary Fig. 4 and Supplementary Table 2). Given the impact of S668A on Fab235 binding, however, it is likely that conformational exchange between H-bonded and unbonded states is functionally significant. Collectively, these structural data agree well with the functional results shown in Fig. 1d.

The $V_H$ domains of *Apo* and *Holo* overlap with an RMSD value of 0.29 Å (Cα atoms only) or 0.48 Å (all atoms), suggesting a typical "lock and key" binding mode (Supplementary Fig. 5). Nonetheless, the V- and C-modules move relative to each other due to a range of adopted elbow angles unrelated to ligation per se. However, as described

below, such flexibility is insufficient to overcome a mismatch between the Fab235 vectorial binding and the MPER-approach angle required to enter the gp160 crawlspace, impacting Fab235 function[10].

To directly compare the MPER-approach angle of Fab235 with MPER-C-specific human bnAbs 4E10 and 10E8, we aligned all three complexes based on their MPER-C helical residues D674 to W680 (Fig. 2c). The coordinate system depicted in Fig. 2d allows a comparative analysis of MPER-approach angles in the trimer context. 4E10 and 10E8 bind from one side of MPER-C ($0° < \phi < 90°$) at about 45° and 30°, respectively, while, in contrast, Fab235 approaches the MPER-C from the other side ($90° < \phi < 180°$) using a more vertical insertion angle, ~110° (Fig. 2c). Interestingly, the protruding Fab235 CDR1$_L$ is positioned to interact with the virion membrane potentially via residues L30$_L$ and Y31$_L$, among others (additional details in Fig. 2 legend). Contacts adjacent to the MPER may stabilize the Fab-MPER interaction employing CDR1$_L$ of Fab235 in lieu of CDR3$_H$ membrane contacts observed previously for bnAbs[13,18–23]. Notwithstanding the paratope-membrane interplay, Fab235 should be largely blocked from binding to the MPER in the spike context (Fig. 2d). As the C helix of the MPER is located more centrally than the N helix relative to the 3-fold axis of the gp160 trimer[10], unfavorable MPER-approach angles may further impede access to the MPER-C.

## Fab460 recognition of a core epitope shared with human bnAb 2F5

Fab460 structures generated from individual crystals using several different conditions are nearly identical, being characterized by relatively flat CDRs (Supplementary Table 1 and Supplementary Fig. 6). The top of Fab460 reveals two positively charged pockets contributed by $V_H$ residues R50$_H$ and K59$_H$ for one and R100$_H$ and R98$_H$ for the other. While $SO_4^{2-}$ anion groups from the crystallization buffer occupy these pockets in the *Apo* structure (Supplementary Fig. 6), the latter could potentially bind acidic MPER residues in the *Holo* structure, a speculation borne out as shown below. The 138.5° elbow angle of Fab460 falls within the most typical range for kappa chain-containing Fabs[35] but with an uncommon salt bridge formed between R41$_L$ from the CC′-loop at the bottom of the $V_L$ domain and E164$_L$ from the DE-loop at the top of the $C_L$ domain (Fig. 3a). This salt bridge may limit rotation between V domains and C domains and/or elbow angle change.

Next, an MPER-N peptide (a.a. D659 to N677) was used for co-crystallization and a complex structure was obtained from a single condition with one Fab460/MPER-N complex per asymmetric unit (Fig. 3a). Despite a nominal 3.5-Å resolution limit, the $V_L$ and $V_H$ domains of Fab460 including their CDRs are well resolved with the MPER-N unambiguously identified and built into the structure (Supplementary Fig. 7 and Supplementary Tables 1 and 3). The two constant domains ($C_L$ and $C_{H1}$) are partially disordered, especially at their base and including the L1, L2 and L3 loops, presumably impacted by a lack of lattice contact. As shown in Fig. 3a, b, the MPER-N peptide binds within the groove between the $V_L$ and $V_H$ domains. There is no significant overall conformational change upon peptide binding aside from a slight widening of the binding groove associated with an increase of the Fab elbow angle from 138.5° to 147.6° (Supplementary Fig. 8), perhaps consequent to different molecular packing within crystals. More importantly, the N-terminal segment of the peptide (D659 to W666), which is largely in a loop conformation ending with a β-turn as discussed below, plays a primary role in the interaction between the MPER-N peptide and Fab460 (Fig. 3b and Supplementary Table 3). The C-terminal peptide segment (A667 to N677), on the other hand, is helical and protrudes away from the Fab-binding groove. The N-terminal part interactions include hydrogen bonds, D659$_{MPER}$ to K59$_H$ (C″ strand) and D664$_{MPER}$ to Y33$_L$ (CDR1$_L$) and to W90$_L$ N and S91$_L$. Both W90$_L$ and S91$_L$ are part of CDR3$_L$. The sidechain of E662$_{MPER}$ points into the positively charged pocket to form a salt bridge with

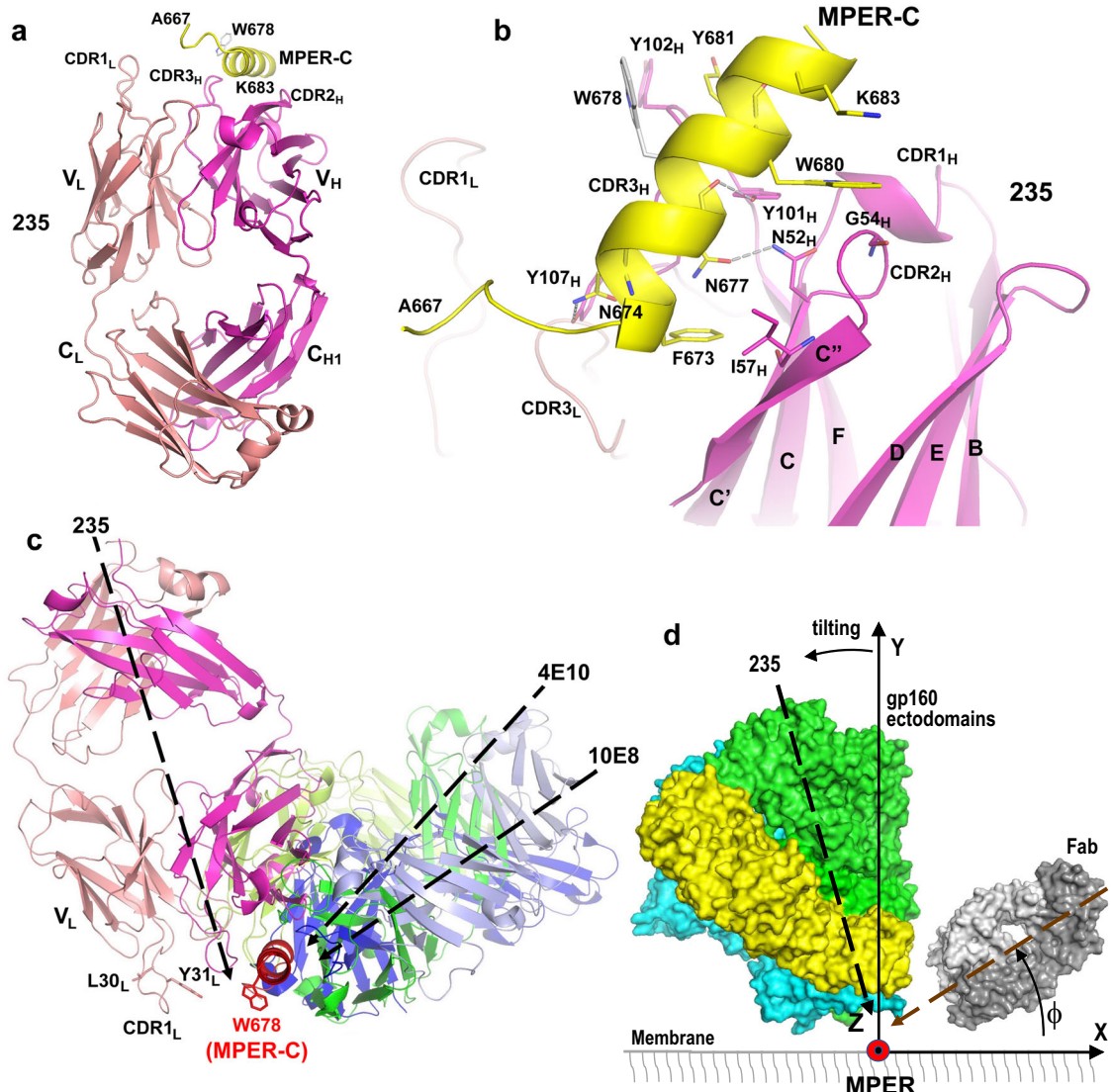

**Fig. 2 | Structural analysis of the Fab235/MPER-C helix complex and the distinct Fab235-MPER-approach angle revealed in comparison to those used by bnAb Fabs. a** Ribbon diagram of Fab235 in complex with MPER-C. MPER residue W678 in gray stick representation indicates the orientation of the MPER-C helix. Heavy chain, light chain and MPER-C are colored in magenta, salmon and yellow, respectively. **b** Interactions between Fab235 and MPER-C. The MPER-C (C atoms in yellow) and major interacting paratope residues from Fab235 (C atoms in magenta) are drawn in stick representation. Hydrogen bonds are shown as gray dashed lines. **c** View of indicated Fabs approach to the MPER-C from an overlay of X-ray structures (PDBs: 2FX7 and 4G6F) aligning MPER D674 to W680 and shown with the MPER largely membrane-immersed. For clarity, residues L30$_L$ and Y31$_L$ on one side of the protruding CDR1$_L$ of Fab235 marked in stick representation denote potential interactions of the CDR1$_L$ loop with the membrane. This loop also includes residues, S$_{32}$SNQK$_{36}$ that may undergo conformational change upon Fab binding to the

MPER. For example, K36 may also participate in interactions with the phosphate head groups of membrane lipids. **d** A graphic depiction, showing the approach of a Fab to one MPER-C protomer following gp160 ectodomain tilting (25°) to the opposite side and granting Fab access to the MPER-C. To compare each antibody approach to the MPER on the virion surface before CD4 engagement, we defined a coordinate system such that the MPER-C helix is positioned on the Z-axis with N- to C-terminus direction parallel to the Z-axis direction. The membrane lies on the *xz*-plane, with the MPER residing in its crawlspace. The Y-axis is perpendicular to the membrane. We use the pseudo-2-fold axis of a Fab as a vector to represent the orientation of the Fab. The insert angle of a Fab is denoted by Φ. Each gp160 protomer is shown in a distinct color. For clarity, the crawlspace is not drawn to scale. The dashed black line represents the relative approach of Fab235 as in panel (**c**) and hence Fab235 is blocked from binding even with tilting.

R50$_H$ positioned immediately proximal to CDR2$_L$. Hydrophobic contacts include those of L663$_{MPER}$ and two residues, I93$_L$ and W90$_L$, on the CDR3$_L$ loop. There are also two mainchain-mainchain hydrogen bonds, the N atom of D664$_{MPER}$ to the O atom of W90$_L$ and the O atom of L661$_{MPER}$ to the N atom of Y102$_H$ (not shown in Fig. 3b). Additionally, W666$_{MPER}$ is sandwiched by the aliphatic component of the K665$_{MPER}$ and L660$_{MPER}$ sidechains, forming a hydrophobic patch, which further stabilizes the conformation of the N-terminal part of MPER-N. There is one additional hydrogen bond from the helix to Fab460, N671$_{MPER}$ to Y102$_H$ in CDR3 (Fig. 3b).

The most interesting feature in the Fab460 *Holo* structure is that the core of the hydrophobic patch is part of a type I β-turn, comprising residues D664$_{MPER}$ to A667$_{MPER}$ (Fig. 3b). The β-turn is the conformation adopted by the MPER segment after Fab binding. This structural motif is also observed in MPER peptide complexed with Fabs 2F5 and m66 (Fig. 3c and Supplementary Figs. 9, 10), reflecting a conserved core epitope[19,36] that is part of a well-structured, larger loop specific to Fab460. The β-turn leads into the N-terminal helix of MPER. The common epitope for Fabs 460, 2F5 and m66 lies between the gp41 C helix (HR2) and the more distal N-terminal MPER, creating an elbow/

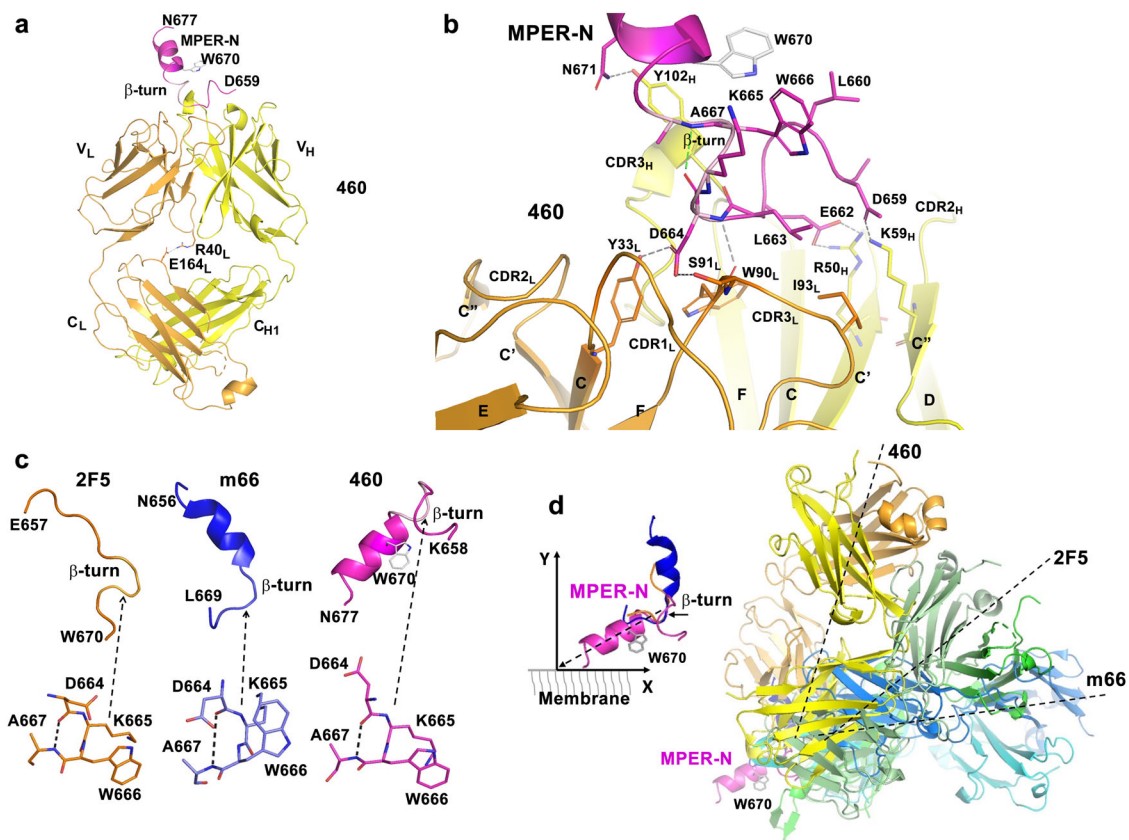

**Fig. 3 | A similar binding mode but a disfavored MPER-approach angle of Fab460 compared to patient-derived Fabs from 2F5 and m66. a** Ribbon diagram of Fab460 in complex with MPER-N. Heavy chain, light chain and MPER-N are colored in yellow, orange and magenta, respectively. MPER residue W670 is drawn in stick representation to indicate the MPER-N helix orientation. The β-turn within MPER-N is displayed in pink. The two residues, $R40_L$ and $E164_L$, that form a salt bridge between the $V_L$ and $C_L$ domains are also highlighted in stick representation. The sidechain of $E164_L$ in the Fab460/MPER-N complex structure is disordered. Its position in the complex shown in the figure is modeled based on the *Apo* crystal structure for the purpose of illustration. **b** The interaction pattern of Fab460 with MPER-N. The MPER-N and major interacting paratope residues from Fab460 are shown in stick representation. Hydrogen bonds and salt bridges are shown as gray dashed lines except for the hydrogen bond in the β-turn, which is shown in green. All β-strands and CDR loops are labeled. **c** Similarity of the β-turns found in the MPER peptides in their respective structures in complex with 2F5, m66 and Fab460. MPER residue W670 is drawn in stick representation to indicate the orientation of the MPER-N helix. **d** A superposition of three Fab/MPER complex structures (on the right) based on their common MPER β-turns. The MPER helical part in the Fab460/MPER-N structure is selected to represent the N-terminal helix of the MPER that dynamically moves off the membrane. The selection of the lifting angle of MPER-N helix (-30°) on the membrane is consistent with cryo-EM data (10). The MPER C-terminal helix is not shown in the figure.

hinge between these two segments. Whether gp160 ectodomain tilting facilitates β-turn propensity prior to Fab-arm ligation is currently unknown. As shown in Fig. 3d (right), however, Fab460 has a greater approach angle (-80°) relative to Fab2F5 (-30°) or Fabm66 (-10°) using the coordinate system defined in Fig. 3d (left). Thus, Fab460 access to the MPER is likely restricted, even with gp160 ectodomain tilting. In addition, although the 13H11 mouse mAb, elicited against the recombinant gp140 trimer recognizes an epitope partially overlapping with that of 2F5, the 13H11 binds a well-defined helical structure, inferred to also bind the post-fusion 6-helix bundle structure[37]. Notwithstanding, it is unlikely that 460 binds to the post-fusion state of a gp41 protomer due to major steric clashes with an adjacent gp41 MPER protomer[38].

## The IgG3 hinge impact on Env binding by vaccine Abs vs human bnAbs

Based on the Fab460 and Fab235 results above, we functionally evaluated the approach angles of other vaccine-elicited rmAbs by characterizing their ability to bind the MPER component of gp145 expressed on the surface of 293T cells. To this end, the MPER-N-specific 275, and the MPER-C-specific 203, 484 and 274 rmAbs (Fig. 1d) were produced and purified. The affinity of those rmAbs for MPER/liposomes ranged from 5.5 nM to 63 nM (Fig. 1d and Supplementary Fig. 11a). While -68% of 293T cells expressing the Env derived from the

ADA strain was stained by 2F5 with a median fluorescence intensity (MFI) value of 4242 at 10 μg/ml as measured by flow cytometry, only 2-8% of cells were weakly stained by vaccine-elicited rmAbs (Fig. 4a). The latter manifest low MFI values ranging from 12 to 138 at 20 μg/ml (Fig. 4a and Supplementary Fig. 11b). The very weak Env staining by the IgG1 isotype rmAbs was not predicated on affinity since that of 484 for the MPER was much higher than that of 4E10 germline as a comparison (5.5 nM vs 154 nM). These data and the crystallographic results above imply that the approach angles of the rmAbs to gp160 are distinct from those of human bnAbs, suggesting a vital role for the steric occlusion of the MPER on the virion surface in impacting protective immunogenicity. Consequently, liposome vaccines that array the MPER alone (i.e., removed from the trimer context) likely elicited a majority of Abs that lack the requisite MPER-approach angle (Fig. 1c).

It has previously been shown through molecular engineering that increased distance and flexibility between Ab combining sites enhance 4E10 bnAb neutralizing activity[39]. Excepting PGZL1 and VRC42.01, all MPER-specific human bnAbs arise from the numerically minor IgG3 isotype[13,17,18]. Notably, the IgG3 hinge between the Ab Fab and Fc ($C_{H2}C_{H3}$) domains consists of 62 amino acids with 11 disulfide bonds compared with the IgG1 hinge that is composed of 15 amino acids and has only two disulfide bonds (Fig. 4b). Moreover, the upper 12 amino-acid hinge of IgG3 lacks a cysteine residue that forms a disulfide bond

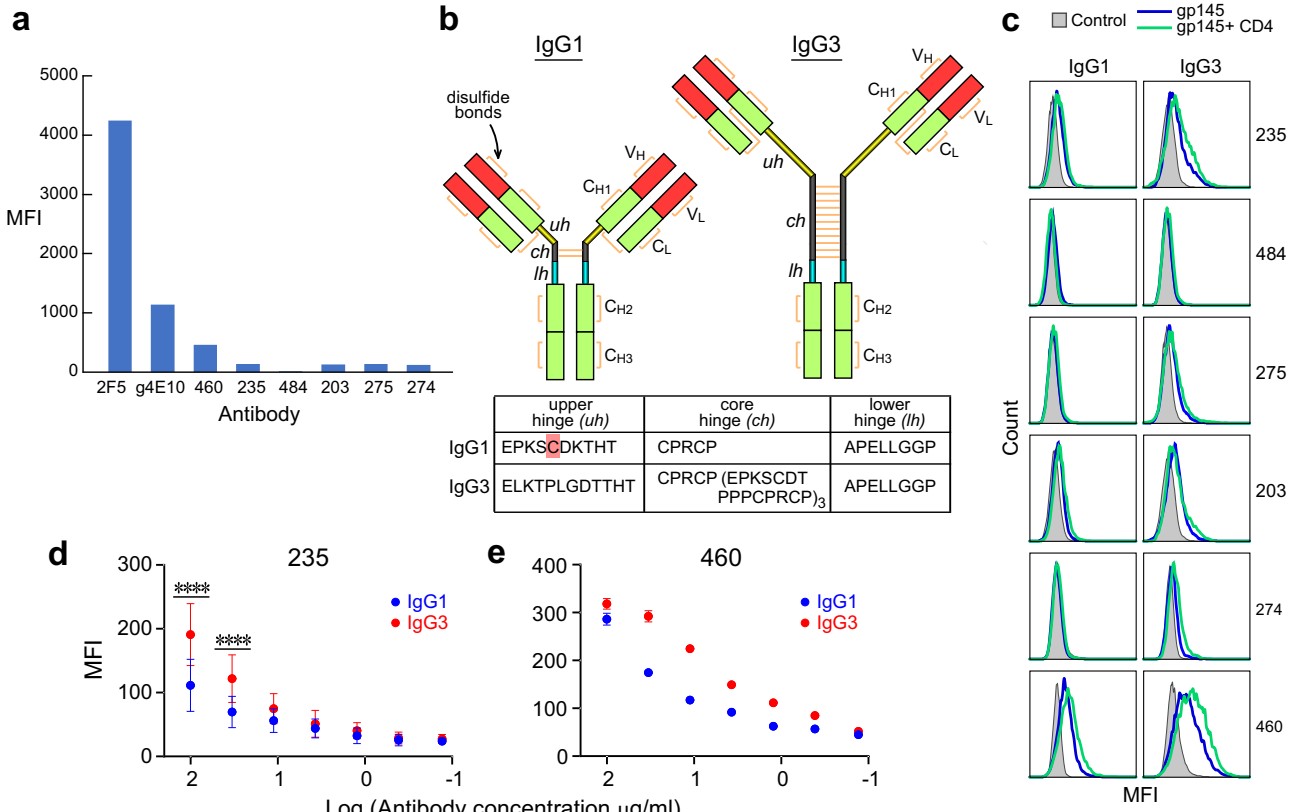

**Fig. 4 | Low Env reactivity of MPER/liposome-elicited IgG1 rmAbs improved by conversion to IgG3 subtype. a** Flow-cytometry analysis of Env binding by vaccine-generated rmAbs. The median fluorescence intensity (MFI) of the indicated 6 rmAbs binding to ADA gp145 expressed on the surface of 293T cells is compared with that of human bnAb 2F5 and germline 4E10 at the concentration of 20 µg/ml for 235, 275, 274, 203, 484 and 460 IgG1, and 10 µg/ml for 2F5 and germline 4E10 IgG1, respectively. **b** Illustration showing the differences in structure of human IgG1 and IgG3 subtypes with tabulation of the length and amino-acid sequence of their hinge regions, with cysteine (C) highlighted that is present in the former but not the latter. **c** Differences in ADA gp145 binding by IgG1 (left) and IgG3 subtypes (right) of vaccine-elicited rmAbs in the presence (green) and absence of soluble CD4 (blue) at 20 µg/ml by flow cytometry. Negative control staining with each corresponding antibody is in gray. Representative of $n = 2$ biologically independent experiments shown in (**a**) and (**c**). **d**, **e** Concentration-dependent ($\log_{10}$) ADA gp145 binding by vaccine-elicited 235 (**d**) and 460 (**e**) as IgG1 versus IgG3 subtypes. Data are represented as mean of $n = 4$ (**d**) and $n = 2$ (**e**) biologically independent experiments. Error bars represent standard deviations. Statistically significant differences between IgG subtypes were determined by 2-way ANOVA, denoted by $p$ values: ****$p < 0.0001$. Source data for **c**–**e** are provided as a Source Data file.

with a $C_L$ residue in IgG1, collectively enabling higher Fab rotational freedom with attendant flexibility, greater span of the two Fab arms relative to each other and separation from the Fc domain to reduce steric hindrance[40,41]. Given these attributes, we assessed whether the IgG3 hinge could influence gp160 binding by the six vaccine-elicited rmAbs, produced as both IgG1 and IgG3 subtypes. As shown in Fig. 4c, Ab staining of cell-surface Env was modestly enhanced by the IgG3 subtype of 235, 275, 203, 274 and 460 at 20 µg/ml compared to their respective IgG1 counterparts, disclosing further enhancement after engagement of soluble CD4 with gp120. This improved Env binding by IgG3 over IgG1 subtype was not accompanied by differences in affinity for the MPER/liposomes as determined by ELISA (Supplementary Fig. 11c). Ab binding dose-response curves exhibited an increase in cell staining by 235 and 460 IgG3 by 20–90% and 28–96% at various concentrations compared to that of IgG1, respectively (Fig. 4d, e). However, neither 484 IgG1 nor 484 IgG3 showed gp160 binding (Fig. 4c).

We observed equivalent ELISA affinity for MPER and a similar cell-surface Env staining for ADA gp145 between IgG1 and IgG3 subtypes of all the various human bnAbs 2F5, 4E10, 10E8 as well as nAbs Z13e1 and m66.6 (Supplementary Fig. 11d, e). In addition, IgG1 and IgG3 subtype binding to ADA gp145 was not influenced by glycans at residue N88 and N625 proximal to the MPER using 10E8 as a proxy for an antibody targeting the MPER-C (Supplementary

Fig. 12)[7]. Thus, neutralizing activity by mature 2F5 and 4E10 was not benefited by the hinge domain of IgG3 (Supplementary Table 4), concordant with previous data reported for the mature bnAbs 2F5, 10E8 and LN01[16,26,27].

### Anti-viral activity of germline bnAbs 2F5 and 4E10 augmented by IgG3 subclass

Given the early rise of IgG3 Abs found in the plasma of HIV-1-infected individuals[42], we next determined whether germline-related (g) MPER-specific bnAb function is more dependent on the IgG subtype. To this end, the V domains of 2F5, 4E10, 10E8, Z13e1 were reverted to the most homologous germline alleles, respectively, without altering the CDR3$_H$ loops (Fig. 5a). Neither IgG1 nor IgG3 subtypes of g10E8 or gZ13e1 were able to stain 293T cells expressing ADA and BG505 Env, respectively (Supplementary Fig. 13a). On the other hand, g2F5 IgG1 showed very weak binding to the Env derived from various strains and a pronounced 3- to 14-fold enhancement of g2F5 IgG3 staining for 293T cells expressing ADA, JR-FL, ZM651 and BG505 gp145, and VC20013 gp160 compared to gIgG1 (Fig. 5b). Similar trends were observed for g4E10 with a 1.5- to 3.5-fold increase in binding by gIgG3. Dose-dependent response curves against ADA gp145 exhibited average 2.7- and 2-fold improvements in Env binding by the gIgG3 of 2F5 and 4E10 compared to that of IgG1, respectively (Supplementary Fig. 13b). Despite binding magnitude differences modulated by strain and epitope-specific

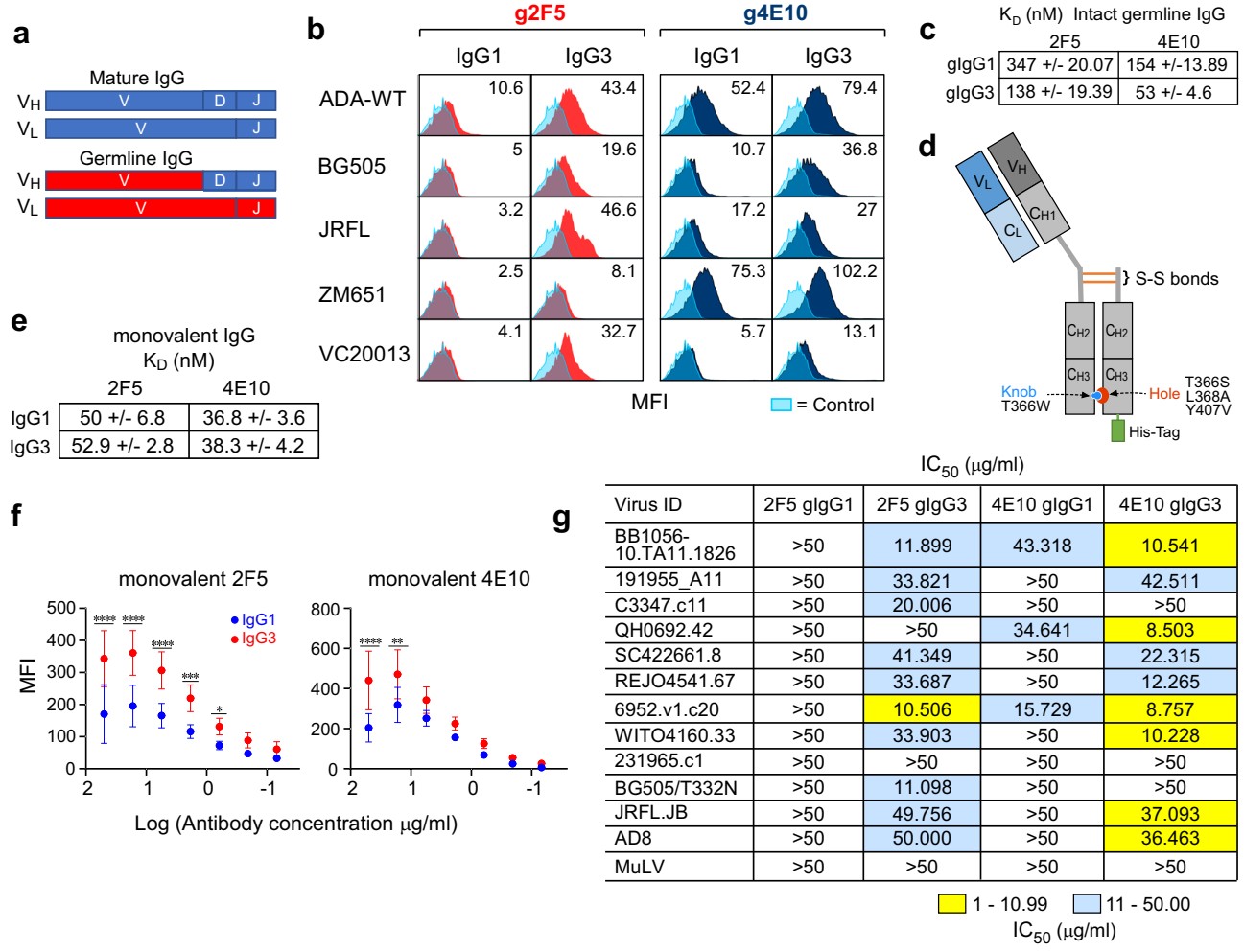

**Fig. 5 | Augmented Env binding, MPER access and anti-viral activity mediated by germline 2F5 and 4E10 IgG3 compared to IgG1 Ab subtype. a** Cartoons of mature and germline anti-MPER bnAbs. V, D, and J segments of mature $V_H$ and V and J segments of mature $V_L$(k) domains are indicated in blue with germline reversions marked in red, creating germline 2F5 (g2F5) and germline 4E10 (g4E10) bnAbs. **b** Histograms with MFI values comparing Env binding between IgG1 and IgG3 subtype of g2F5 (red) and g4E10 (dark blue) by flow cytometry with negative controls in light blue. Antibodies at 10 μg/ml were tested for binding to Envs derived from the various indicated HIV-1 strains. The results shown are representative of $n = 2$ independent experiments performed. **c** $K_D$ of intact g2F5 and g4E10 measured by SPR using an L1 chip for capturing MPER/liposome complexes assembled using a peptide: lipid ratio of 1:1000. **d** Schematic of the design of a one-armed monovalent antibody using IgG1 as an example. The heavy chain

incorporates a knob mutation, and the truncated Fc domain incorporates hole mutations as indicated and described previously[43]. **e** $K_D$ values for binding of monovalent IgG1 and IgG3 subtypes of mature 2F5 and 4E10 to MPER/liposome complex by SPR analysis. **f** Comparison of dose-dependent ADA gp145 binding by flow cytometry of monovalent IgG1 versus monovalent IgG3 subtypes of mature 2F5 and 4E10 according to log$_{10}$ dilutions shown. Data are represented as mean of $n = 3$ biologically independent experiments. Error bars represent standard deviations. Statistically significant differences between IgG subtypes were determined by 2-way ANOVA, denoted by $p$ values: ****$p < 0.0001$; ***$p$-value of 0.0003; **$p$-value of 0.0018; *$p$-value of 0.0284. **g** Neutralization profile for IgG1 and IgG3 subtypes of g2F5 and g4E10 against HIV-1 Env pseudoviruses. Source data for (**b**), (**c**), (**e**), and (**f**) are provided as a Source Data file.

differences, the findings collectively reveal better binding by gIgG3 to the MPER of a variety of Env strains.

We assessed whether the potentiated IgG3 Env binding tracked with stronger Ab binding avidity. The relative binding avidities of IgG1 and IgG3 subtypes of g2F5 and g4E10 were measured by ELISA for MPER/liposomes at high and low peptide-to-lipid ratios (1:50 vs 1:500, mol:mol). In contrast to the mature 2F5 and 4E10 bnAbs, the EC$_{50}$ values indicated a requirement for a > 14-fold higher concentration of IgG1 compared to that of the IgG3 version of g2F5 and g4E10, when the MPER density is high on the liposome membrane surface. The differences in EC$_{50}$ values were even more pronounced (28-fold) with lower MPER density (Supplementary Fig. 13c), underscoring the influence of antigen density on the binding avidity of the germline bnAbs. Similarly, $K_D$ values of g2F5 and g4E10 of the IgG3 subtype were 2.5- to 3-fold better than that of the IgG1 subtype when tested by SPR against MPER/liposome at a 1:1000 ratio (Fig. 5c and Supplementary Fig. 13d).

To assess whether the augmented gp160 binding by the IgG3 versions of g2F5 and g4E10 was facilitated through greater accessibility to the sterically restricted MPER embedded in Env, monovalent IgG1 and IgG3 versions of bnAbs 2F5 and 4E10 were produced. This eliminated Ab avidity as a contributor to epitope-binding strength. For this purpose, we incorporated knob and hole mutations in the constant heavy chain domain 3 ($C_{H3}$) of the Ab Fc, engineering IgG molecules with a single Fab arm (Fig. 5d)[43]. These mutations promote heterodimerization of the hole-containing Fc domain and the knob-containing heavy chain, facilitating monovalent Ab assembly (Supplementary Fig. 13e). Similar affinity between monovalent mature 2F5 and 4E10 IgG1 and IgG3 was exhibited in SPR against MPER/liposomes (Fig. 5e and Supplementary Fig. 13f). The binding of monovalent g2F5 and g4E10 to the MPER on the Env-expressing 293T cell surface was too weak to be detected by FACS (Supplementary Fig. 13g). On the other hand, both mature 2F5 and 4E10 monovalent

IgG3 exhibited a 1.2-to-2-fold improvement in ADA gp145 binding by FACS over the corresponding monovalent IgG1 at all concentrations tested (Fig. 5f).

Next, we determined if the greater IgG3 variant binding observed above impacted neutralizing activity of intact bivalent g2F5 and g4E10 using a panel of 12 viruses in the TZM-bl assay. Whereas no neutralizing activity against the tested viruses was observed for g2F5 IgG1, g2F5 IgG3 neutralized 10 of the 12 viruses at $IC_{50}$ values ranging 10–50 μg/ml. The neutralization potency and breadth were also improved for g4E10 IgG3 versus IgG1 (Fig. 5g). The augmented anti-viral function of intact g2F5 and g4E10 IgG3 is likely to be a consequence of both increased MPER accessibility and avidity mediated by this Ab isotype. Contrary to the influence of the IgG3 hinge on the function of germline bnAbs, the monovalent mature bnAbs manifest equivalent IgG1 and IgG3 neutralization activity across the 12 tested HIV-1 viruses except for a several-fold improvement in $IC_{50}$ against the BG505/T332N and C3347.c1 viruses by the monovalent 2F5 and 4E10, respectively (Supplementary Table 5). Overall, for germline bnAbs, the biological function of the anti-MPER IgG3 subtype appears to be significantly improved.

## Inter-spike distance distribution on virions and IgG3 mitigation

To better understand the role of the different hinge lengths of IgG1 and IgG3 subtypes on neutralization, computational modeling was performed. The two hinges of the subtypes form the same ~80° angle and, therefore, the Fab domains were made to orient in an alike manner. This gives a 99-Å IgG1 span, while IgG3 has a 166-Å span (Fig. 6a). These distances were used to examine the capacity of the two subtypes to crosslink spikes (inter-spike) on a virion, as intra-spike crosslinking by anti-MPER bnAbs is known to be unlikely[7,10]. We constructed models of virions with a given number of spikes ($N$) randomly distributed on the virion surface. Based on prior cryo-EM studies[9], an average of 14 trimers of HIV-1 ($N = 14$) with a virion diameter of 110 nm was selected for comparison with $N = 73$ of simian immunodeficiency virus (SIV) mac239 variant virus, a genetically related virion of similar size. The number of spike pairs that have a surface-to-surface distance of less than 99-Å or 166-Å is denoted as $L_{99}$ and $L_{166}$, respectively (Fig. 6b), and the distributions of $L_{99}$ and $L_{166}$ are measured from $10^5$ models generated for $N = 14$ and $2 \times 10^4$ for $N = 73$, respectively (Fig. 6c, d). With sparse trimer density ($N = 14$) on HIV-1, there is an average $2.3 \pm 1.4$ (avg ± sd) spike pairs within a distance of 99 Å and $4.7 \pm 1.9$ pairs within 166 Å. On the other hand, at the greater density of 73 trimers found on SIV there are $70.7 \pm 5.4$ pairs of spikes that are less than 99 Å apart and $137.8 \pm 6.7$ that are less than 166 Å.

For examination of the percentage of spikes that can be crosslinked, we calculated nearest-neighbor distance distributions (Supplementary Fig. 14). For $N = 14$, 23% of the spikes have nearest-neighbor distance less than 10 nm, while for $N = 73$, the percentage becomes 90%. Thus, most of the spikes can be crosslinked when $N = 73$ so that the separation of peaks in Fig. 6d does not indicate that IgG3 is more effective than IgG1 in the SIV case (also see Supplementary Fig. 14). Using different models where IgG1 is even more fully extended while IgG3 less so, shows a similar trend (Supplementary Fig. 15). It is noteworthy that these comparisons do not consider orientational flexibility of Fab domains, where Fabs of IgG1 have less rotational freedom, and hence their capacity to crosslink between spikes will diminish further. Thus, the impact of IgG3 on avidity may trend inversely with viral spike density, suggesting that the biological function of IgG3 would be greater when the density of the spikes on a virion is low.

## Discussion

The quaternary structure of an epitope and its array density on the virus' surface impact Ab protective function. For hemagglutinin (HA) of influenza A viruses, for example, significant affinity improvement via the bivalent binding of Ab appears to be particularly pertinent for heterotypic neutralizing activity[44]. However, the extent of avid binding is also an epitope- and structure-dependent property, as different Abs vary in the ability to crosslink their antigens. In the case of HIV-1, both the distance between spikes being greater than the reach of the two Fab arms of one immunoglobulin molecule and the geometric constraint imposed by epitope orientation disfavor bivalent attachment, instead fostering monovalent IgG binding[45]. Thus, avidity plays a relatively minor role in Ab affinity for the viral spike and protective potency against HIV-1, as evidenced previously for bnAbs against gp120 and, in particular, the MPER for which there is sterically limited access to epitopes[46,47]. Furthermore, the benefit of avidity is marginal when Fab affinity is high[48]. Since the binding strength of mature 2F5 and 4E10 Fab is near maximal, small improvements in the MPER accessibility and/or avidity mediated by the IgG3 subtype compared to those of IgG1 may not further increase neutralization activity (Fig. 5f and Supplementary Tables 4 and 5). By contrast, the impact of structural flexibility intrinsic to the IgG3 subtype on avidity and consequent protective function is more pronounced on germline version of 2F5 and 4E10 bnAbs. Notable in this context is the observations that Env trimers (-5–7) cluster on HIV-1 virions to form a so-called entry claw in the contact zone between the virus and the target cell[9,49,50]. Given a small number of trimers required for virus fusion and sequence variations between diverse HIV-1 strains/isolates, the reduced bnAb binding affinity for heterologous strains can be compensated by enhanced inter-spike crosslinking by IgG3 (Fig. 6c and Supplementary Fig. 15c). The 2-fold increment in spike crosslinking without considering the Fab orientational flexibility (Fig. 6c) should be a lower bound, which is significant for germline bnAb IgG3 (Fig. 5c, g and Supplementary Fig. 13b, c) to disrupt cooperative action of several such trimers required for viral membrane fusion and/or reduce the probability of recruiting the requisite number of trimers to the contact zone[51]. In that context, the biological function of IgG3 would be greater when the density of the spikes on a virion is low such as for HIV-1 and SARS-CoV2 (24-26 trimers)[52,53]. Conversely, the benefit of IgG3 may be more limited in the case of influenza virus (300–500 spikes) as evidenced against antigenically matched viral strains[54–56]. Note that most of the spikes can be crosslinked when $N = 73$ (Fig. 6 and Supplementary Fig. 15). Notwithstanding, greater neutralization capacity of IgG3 versus IgG1 subtype was recently observed against antigenically drifted influenza virus presumably due to concomitant reduction in Ab affinity resulting from mutation[56].

A recent longitudinal study of the neutralizing Ab CAP88-CH06 lineage directed against the C3 region of gp120 showed that IgG3 and IgA isotypes are better able to neutralize longitudinal autologous viruses and their variants than IgG1[57]. Consequently, the biological importance of IgG3, given its intrinsic hinge and isotype geometry impacting Fab flexibility, Fab-Fab arm distance and Fc functionality is not restricted to the MPER. MPER-specific B cells and those directed at other specificities may benefit from utilizing the IgG3 isotype as a mean to achieve germinal center competitiveness to mitigate intrinsically weak affinity, obscured accessibility and/or viral mutations in early immune responses. In the case of HIV-1, subsequent somatic hypermutation would shape the paratope site to augment affinity, optimize the vectorial approach to the MPER and increase the Fab elbow angle and/or otherwise improve Fab dynamics to foster entry into the confined MPER crawlspace of the HIV-1 Env.

Certain epitope targets of bnAbs are often only transiently exposed during dynamic motions of the Env trimer or during the viral fusion process[10,58]. Sterically limited epitopes such as those on the anchor epitope of the influenza HA stalk, the coronavirus glycoprotein S2, and the MPER of the Ebola virus glycoprotein are immunologically subdominant in response to natural infection or vaccination[59–62]. A challenging angle for bnAb approach to the HR2-MPER of Ebola virus glycoprotein SP2 has also been revealed[59]. Collectively, the cryptic nature and conserved linear sequence of those target epitopes merit

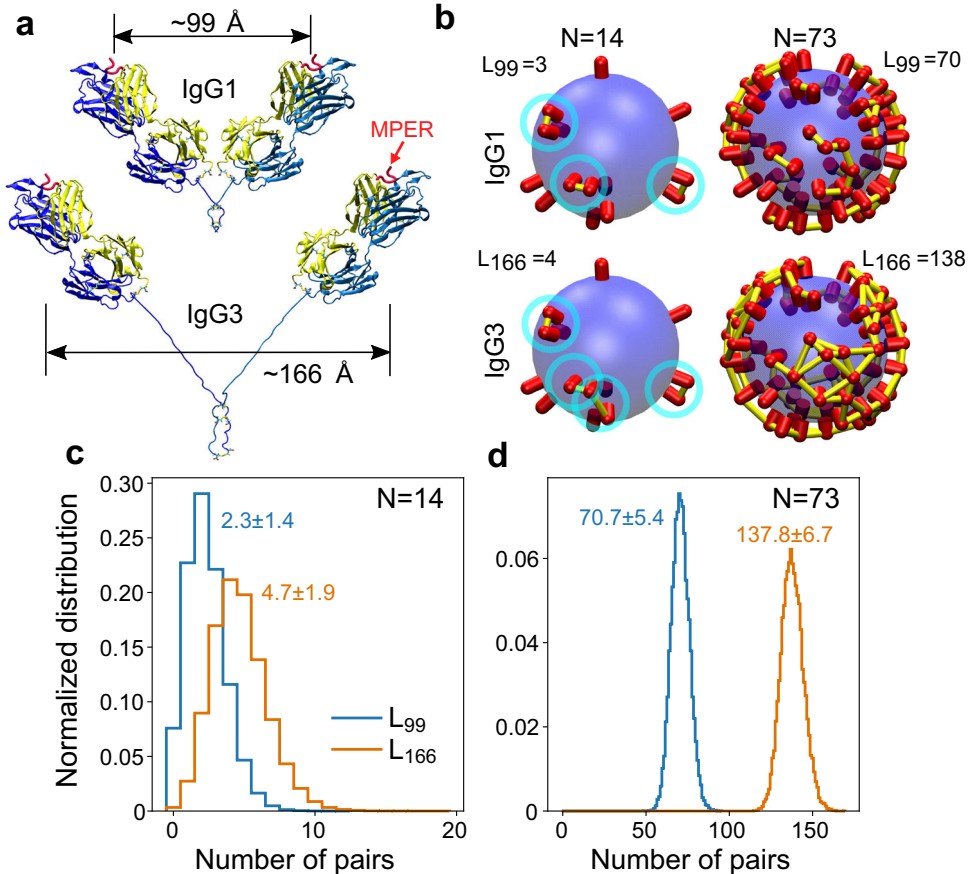

**Fig. 6 | Comparison between IgG1 and IgG3 in their potential to form inter-spike crosslinks. a** Structural models of IgG1 and IgG3. PDB 1TJG[19] was used to add the Fab 2F5 domains. Upper hinges and Fabs have the same orientations between the two subtypes. Supplementary Fig. 15a shows models with wider spans between the two Fab domains, in particular IgG1. **b** Examples of models of virions with 14 and 73 randomly placed spikes to represent HIV-1 and SIV, respectively. The number of spike pairs that have a surface-to-surface distance of less than 99-Å or 166-Å is denoted as $L_{99}$ and $L_{166}$, respectively. Yellow bars (highlighted by cyan circles for $N = 14$) connect between spikes that are less than 99 Å and 166 Å apart. Numbers of spike pairs within these distance cutoffs are shown in each panel. **c, d** Distributions of number of pairs within cutoff distances as indicated for $N = 14$ (**c**) and $N = 73$ (**d**). In each case, $10^5$ and $2 \times 10^4$ models were generated with randomly placed spikes. Numbers are avg ± sd in each case. Distributions of nearest-neighbor distances for individual spikes are shown in Supplementary Fig. 14.

focused immunogen design strategies. Without vectorial constraints to select Abs with a requisite approach angle, however, induction of protective high-titer serum Abs such as anti-MPER bnAbs can be challenging (Fig. 1c).

While no neutralizing activity was detected in immune sera from individuals vaccinated with MPER/liposomes in the HTVN133 trial[63], consistent with our murine studies, a fraction of peripheral blood B-cell-derived rmAbs was able to weakly neutralize tier-2 viruses, demonstrating a proof of concept for vaccine elicitation of bnAbs specificity directed against the MPER in humans. Although we can't exclude the possibility that the lack of nAb induced by MPER/liposome vaccine in the murine model may be impacted by the relatively short mouse CDRH3 lengths compared to those of the human[13,14], clearly, this trial emphasizes the necessity of improving MPER/liposome immunogen design and/or immunization strategy to elicit sufficient plasma neutralizing Abs affording protection against subsequent HIV-1 exposures.

Our results advocate a broadly applicable vaccination strategy: amalgamating a quaternary structural mimic with a linear target epitope for immunogen design to select bnAbs against high-value antigenic sites that are obscured by a virus for its own protection. RNA-based vaccines for HIV-1 comparable to those that performed effectively for COVID-19[64] will require substantial modifications, including removal of misguiding viral epitopes that defocus responses against the cryptic and conserved MPER. In addition, the benefit of IgG3 isotype elicitation revealed here, including for Ab generation that precedes somatic hypermutation and affinity maturation, should facilitate a protective vaccine response.

## Methods

### Mice and MPER/liposome immunizations

BALB/c mice were obtained from Taconic Biosciences. All female mice used were 8–10 weeks of age at the time of initial immunization. Mice were housed in a specific pathogen-free facility and maintained in accordance with procedures and protocols approved by the Dana-Farber Cancer Institute and Harvard Medical School Animal Care and Use Committee Institutional Review Board. Liposome vaccines were made by drying the following components under a nitrogen stream and placing them under vacuum overnight: N-terminally palmitoylated MPERTM peptides (pMPERTM), monophosphoryl lipid A (MPLA) from *Salmonella enterica* serotype Minnesota (Sigma-Aldrich, 6895), and the lipids 1,2-dioleoyl-sn-glycero-3-phosphocholine (DOPC) (Avanti Polar Lipids Inc., 850375), 1,2-dimyristoyl-sn-glycero-3-phosphocholine (DMPC) (Avanti Polar Lipids Inc.850345), 1,2-dioleoyl-sn-glycero-3-phospho-(1'-rac-glycerol) (DOPG) (Avanti Polar Lipids Inc., 840475) and 1,2-distearoyl-*sn*-glycero-3-phosphoethanolamine-N-[methoxy(polyethylene glycol)-2000] (DSPE-PEG) (Avanti Polar Lipids Inc., 880120) at a molar ratio of 2:2:1:1. Lipids were hydrated with phosphate-buffered saline (PBS) pH 7.4 containing 1 mg/ml LACK to a final concentration of 26.5 mM lipid. The LACK156-173 peptide, a well characterized immunodominant CD4 T cell epitope presented by the

I-A$^d$ (MHC class II) molecule, was derived from the Leishmania major RACK-like homolog of the WD protein family[65]. Final liposomes incorporated pMPERTM at a molar MPER:lipid ratio of 1:200 and contained 175 μg/ml MPLA, with a total lipid concentration of 25 mg/ml. MPER/liposomes were sized by vortexing 6 times for 30 s each at 5-min intervals, 6 rounds of flash freezing in liquid nitrogen and thawing at 37 °C, and extrusion by passage through a 200-nm-pore-size polycarbonate membrane (Whatman Inc, WHA10417004) 21 times. Mice were immunized intradermally with 50 μl per hind flank. Immunizations were administered on three occasions at 21-day intervals and the mice were sacrificed 30 days or 100 days after the final booster immunization to assess immunogenicity or to screen MPER-specific plasma cells from bone marrow.

## Peptide synthesis

MPER peptides were generated on an ABI 431 peptide synthesizer by using Fmoc chemistry, high-pressure liquid chromatography (HPLC) purification, and post-purification conjugation of N-terminal palmitic acid at the Massachusetts Institute of Technology. LACK (ICFSPSLEH-PIVVSGSWD) for CD4 T cell epitope peptides, MPER-N (KDLLELDK-WASLWNWFNITNK) and MPER-C (KASLWNWFNITNWLWYIKKK) peptides for crystallization, and Npalm-MPERTM (N-terminally palmitoylated DLLELDKWASLWNWFNITNWLWYIKLFIMIVGGLVGLRIVFA-VLSIVKRVR) for immunogenicity and MPER-specific single B cells analysis, and Npalm-MPER (N-terminally palmitoylated DLLELDK-WASLWNWFNITNWLWYIK) for ELISA, and MPER peptide (ELDK-WASLWNWFNITNWLWYIK) and its alanine mutants were synthesized for epitope mapping analysis.

## Monoclonal antibody engineering, production and purification

The heavy and light chain variable regions of bnAbs of interest were cloned into both IgG1 and IgG3 expression vectors. To generate monovalent antibodies, a knob mutation (T366W) in the CH$_3$ domain of the heavy chain of IgG1 and IgG3 was introduced. Separately, a truncated Fc domain with hole mutations (T366S, L368A, Y407V) along with 6 his-tag fused at the C-terminus of the CH$_3$ domain in IgG1 and IgG3 was constructed by PCR, respectively, to favor hetero-dimerization between two distinct Fc domains[43]. A C220S mutation was also introduced to the upper hinge of IgG1 to avoid potential disulfide bond formed with the light chain of IgG1. The PCR products were cloned into IgG1 and IgG3 heavy chain expression vector, respectively.

mAbs were expressed in suspension Expi293F cells (Fisher Scientific, A14527) cultured in Expi293 expression medium (Fisher Scientific, A1435101). For bivalent IgG, heavy chain (HC)- and light chain (LC)-expressing plasmids were co-transfected in a 1:1 ratio. To express monovalent IgG, the HC, the LC, and the truncated Fc domain were co-transfected in a 1:1:2 ratio for IgG1 and in a 1:1:1 ratio for IgG3, respectively. Co-transfection of those plasmids were performed at a density of 4–5 million cells/mL in a 1:3 ratio (weight: weight) of plasmids to PEImax at 1 mg/ml (Polysciences, 24765). Following a 4-day incubation, transfection mixtures were pelleted by centrifugation and filtered through 0.22 μm Stericup filter units (EMD Millipore, SCGVU05RE). Filtered supernatant was applied to a column containing Gammabind Plus Sepharose (Cytiva, 17088602) equilibrated with phosphate-buffered saline (PBS). The antibodies were eluted with 0.5 M acetic acid pH 3.0 and collected in a 1:5 volume of 3 M Tris pH 9.0. The purified antibodies were buffer exchanged to PBS using 50,000 MWCO Amicon ultra centrifugal filters (Millipore, UFC805024). The IgG concentration was determined on the NanoDrop 2000 and the antibodies were aliquoted and stored at −80 °C until further analyses. The Fab domain was expressed in Expi293F cells and purified by IgG-CH1 Affinity Matrix (Thermo Scientific, 194320005) according to the manufacturer's instructions, followed by superdex 75 gel filtration column (Cytiva). Fab purity was confirmed by 10% non-reduced and 12% reduced SDS-PAGE. Monovalent IgG was further purified by Ni- nitrilotriacetic acid (Ni-NTA) (Qiagen, 30210) column and then size exclusion column (superdex 200) (Cytiva) to remove bivalent IgG.

## Crystallization

Each Fab was concentrated to about 10–20 mg/ml before crystallization. For co-crystallization, the Fab was mixed with the MPER peptide solubilized with 5% DMSO in protein buffer. With a Mosquito nanoliter liquid handler (TTP LabTech), crystallization condition screening was set up using a sitting drop vapor diffusion method with several 96-condition screens, including MCSG1-4, Index, SaltRx, PEG/Ion, PEGs II and Top96 at 16 °C.

Fab235 and F235/MPER-C complex: Fab235 was concentrated to about 17.7 mg/ml. Crystals of Fab235 complex with MPER-C (KASLWNWFNITNWLWYIKKK) appeared under the condition containing 1% w/v Tryptone, 0.05 M HEPES sodium pH 7.0, 12% (w/v) PEG 3350. Crystals of the apo form of Fab235 without MPER-C binding appeared under the condition containing 2 M ammonium sulfate, 0.1 M HEPES: NaOH, pH 7.5, 2 % (v/v) PEG 400.

Fab460 and Fab460/MPER-N complex: Crystallization of apo and holo forms of Fab460 were obtained from different crystallization settings. For the Fab460/MPER-N complex, Fab460 was concentrated to about 10.1 mg/ml. Crystals of Fab460 complex with MPER-N (KDLLELDKWASLWNWFNITNK) appeared under the condition containing 0.2 M ammonium phosphate monobasic, 0.1 M Tris pH 8.5 and 50% v/v (+/-)-2-Methyl-2,4-pentanediol. For Fab460 apo form, the concentration for crystallization was 18.7 mg/ml. Crystals of the apo form of Fab460 appeared under multiple conditions, including the one containing 1.5 M lithium sulfate monohydrate, 0.1 M sodium acetate pH 4.6.

All crystals were harvested and treated with a cryoprotectant solution (25% glycerol in its mother liquor) and then flash frozen in liquid nitrogen before data collection.

## X-ray diffraction and structure determination

X-ray diffraction data were collected at 100 K from cryocooled crystals at the 19-ID beamline of the Structural Biology Center at the Advanced Photon Source at Argonne National Laboratory[66]. HKL3000 program suite was used for data processing, including intensity integration, scaling and merging (Supplementary Table 1)[67]. Structures were determined by using molecular replacement method (Table S1)[68]. The final models were refined using the program Phenix.refine (Supplementary Table 1)[69]. Structural validation of each model was performed using the program MolProbity[70].

Atomic coordinates and structure factors for the reported crystal structures have been deposited in the Protein Data Bank under accession numbers: 8FWF, 8FYM, 8FXJ and 8FZ2.

## Surface plasmon resonance (SPR) measurements

For epitope mapping of vaccine-elicited antibodies, the DOPC/DOPG liposomes in a 4:1 ratio (Avanti Polar Lipids) (30 μl, 150–250 μM) in running buffer (PBS) were applied at a flow rate of 5 μl/min and captured on the Pioneer L1 sensor chip surface (Cytiva) in a BIAcore 3000 instrument. To remove any multilamellar structures from the lipid surface, sodium hydroxide (20 μl, 25 mM) was injected at a flow rate of 100 μl/min, which resulted in a stable baseline corresponding to the immobilized liposome bilayer membrane with response units (RU) of 4000–5000. MPER variant peptide solutions (1–5 μM) were prepared by dissolving in running buffer right before injection and the solution (80 μl) was injected over the lipid surface at a flow rate of 10 μl/min. Antibody solution (10–20 μg/ml) was passed over peptide-liposome complex for 3 min at a flow rate of 10 μl/min. The immobilized liposomes were completely removed with an injection of 40 mM 3-[(3-

cholamidopropyl)-dimethylammonio]-1-propanesulfonate (CHAPS) (Sigma-Aldrich, 331717-45-4) (20 µl) at a flow rate of 5 µl/min, followed by 10 µl injection of NaOH (50 mM)/isopropanol (6:4) at a 20 µl/min flow rate, and each peptide injection was performed on a freshly prepared liposome surface. For $K_D$ measurement of various human bnAbs and vaccine-elicited antibodies, a N-terminally palmitoylated MPER (Npalm-MPER) peptide (DLLELDKWASLWNWFNITNWLWYIK)/liposome complex at 1–1000 ratio of peptide to lipid (mol: mol) was captured on the L1 sensor chip surface with RU ranging from 2500 to 3500. Antibody solution as an analyte was then passed over the Npalm-MPER/liposome surface at the flow rate of 30 µl/min. All interactions were run with an association time of 5 min and a dissociation time of 5 min followed by regeneration of surface of the chip with CHAPS and NaOH (50 mM)/isopropanol (6:4) as described above. Irrelevant rmAb 15E8 specific to SIV gp120 was used as a negative control for subtraction. All binding interactions were fit to a simple 1:1 Langmuir binding model with baseline drift to determine the apparent association ($k_a$) and dissociation rate constants ($k_d$) and equilibrium dissociation constant ($K_D$).

## Flow-cytometry assay

HEK293T cells (ATCC, CRL-3216) at 60–70% confluence were transiently transfected with HIV-1 ADA gp145 Env using polyethylenimine (PEI Max, Polysciences) in a 1:3 ratio of plasmid to PEI (weight to weight). Cells were harvested 48 hours post-transfection in FACS buffer (1x PBS containing 2% FBS, 1 mM EDTA and 0.1% sodium azide). 100,000 cells were incubated with or without 5 µg human soluble CD4 (4 domain) (NIH HIV Reagent Program) at room temperature for 1 hour and then 2 µg vaccine-elicited primary antibody was added to the mixture for another hour of incubation. Germline bnAbs binding to various HIV-1 envelop proteins including ADA, BG505, JR-FL, ZM651 gp145 and vc20013 gp160 was evaluated at the concentration of 10 µg/ml for 1 h at room temperature. For binding kinetics analysis, 100 µl/well of mature monovalent bnAbs starting at 50 µg/ml or of germline bnAbs and of vaccine-elicited rmAbs starting at 100 µg/ml in 3-fold serial dilutions was added to cells and incubated for 1 h. After washing, cells were stained with Phycoerythrin (PE)-conjugated goat anti-human IgG antibody (Southern Biotech, 2040-09, 1:500) mixed with Zombie Aqua Fixable viability dye (BioLegend, 423101, 1:500) at 4 °C for 30 min. Cell-surface fluorescence was analyzed by BD LSR Fortessa. For each transfected cells, the mean fluorescence intensity (MFI) was calculated by subtracting the signal from untransfected cells stained with the corresponding antibody, respectively.

## Enzyme-linked immunosorbent assay (ELISA)

To assess a relative MPER-specific binding reactivity of vaccine-elicited and human bnAbs, 96-well Immulon 2HB plates (Thermo Scientific, 3455) were coated with either 50 µl of DOPC/DOPG (4:1) arraying Npalm-MPER at 1:50 or 1:500 ratio of peptide to lipid at 100 µg/ml of liposome in PBS or 50 µl of streptavidin in PBS at 2 µg/ml overnight at 4 °C. The following day, plates were washed three times with 0.1% BSA-PBS and blocked with 100 µl per well 1% BSA-PBS for 6–8 h at 4 °C or 4 h at room temperature for the N-palm-MPER/liposome- coated plates or for the streptavidin-coated plates. The streptavidin-coated plates were further incubated with biotin-MPER dissolved in DMSO and diluted to 2 µg/ml in PBS right before incubation for 4 h. After washing, serially diluted mAbs in 1% BSA-PBS were incubated overnight with gentle rocking at 4 °C. The following day, goat anti-human–horseradish peroxidase (HRP) (Bio-Rad, 1721050, 1:3000) or goat anti-mouse HRP (Bio-Rad, 1706516, 1:2000) secondary antibody was applied for 1 h at 4 °C. Plates were washed two times with 0.1% BSA-PBS and two times with PBS. Bound antibody was detected by incubation with o-phenylenediamine (OPD) (Sigma-Aldrich, p9029) solution in citrate buffer, pH 4.5, for 7 min. The OPD reaction was stopped with 2.25 M $H_2SO_4$, and the absorbance was read at 490 nm on a Victor X4 plate reader (Perkin-Elmer).

## Neutralization assay

Purified mAbs were tested in duplicate in 96-well plates using a primary concentration of 50 µg/ml and serially diluted 3-fold seven times. HIV-1 Env pseudovirus was added to antibody serial dilutions and plates were incubated for 1 h at 37 °C. TZM.bl cells were then added at $1 \times 10^4$/well with DEAE-Dextran at a final concentration of 11 µg/ml. After 48 h incubation at 37 °C, plates were harvested using Promega Bright-Glo luciferase (Madison, WI) and luminescence detected using a Promega GloMax Navigator luminometer. Antibody concentrations that inhibited 50% or 80% of viral infection were determined ($IC_{50}$ and $IC_{80}$ titers, respectively). Neutralization assays were conducted in a laboratory meeting Good Clinical Laboratory Practice (GCLP) quality assurance criteria.

## Building models of IgG1 and IgG3

Models in Fig. 6a were created by building the Fab and hinge domains separately and joining them. For Fab, PDB 1TJG[19] (https://www.ncbi.nlm.nih.gov/Structure/pdb/1TJG) was used. Hinge sequences cover up to the first 3 disulfide bonds in the core hinge region shown in Fig. 4b. Construction and manipulation of structures were done using CHARMM[71].

IgG1: For the hinge, the 10-residue sequence DKTHTCPPCP was used. Two peptides built for the hinge were placed such that two disulfide bonds can form between them. After building the dimer of hinge peptides, a brief energy minimization was performed to relax the structure. In this model, the two upper hinges of the dimer form about 80° angle. Using the vectors from C241 (first disulfide in the hinge) to C235 (C-terminal disulfide of the Fab CH domain), and from C235 to the Cα-based center of mass of the MPER, the Fab domains of IgG1 were oriented as shown in Fig. 6a. We also created another model where the two Fab domains form a nearly 180° angle, to estimate their maximum span (Supplementary Fig. 15a).

IgG3: Due to its longer length, the hinge was built in two steps, to form disulfide bonds in a sequential manner. First, two 16-residue peptides with sequence ELKTPLGDTTHTCPRC were built and aligned to form a disulfide bond between the first cysteine on the 13th position. Then two 22-residue peptides with sequence ELKTPLGDTTHTCPRC-PEPKSC were built. Coordinates of the first 16-residue peptide dimer was copied to the corresponding parts of the 22-residue dimer, which brings together the remaining 6 residue (aa 17-22) of the dimer in close proximity. The second disulfide bond was then formed for cysteines on the 16th position. A brief energy minimization was performed to relax the structure. Finally, the third disulfide bond was formed for the 22nd cysteine near the C-terminus of the hinge peptide dimer, followed by another brief energy minimization. To build the structure shown in Fig. 6a, we aligned the hinge domain of IgG1 to the IgG3 hinge as built above, using the first disulfides (alignment was done based on Cα atoms). This provides the guiding direction of the upper hinge arms, which we extended to locate a position approximately proportional to the ratio between the lengths of upper hinges in the two subtypes. We then placed the two Fabs so that the Cα atom of K237 is on that position. The orientations of the Fabs were kept the same as in IgG1. The upper hinge region connecting between K237 and T246 was then built using MODLOOP[72].

To build the IgG3 structure in Supplementary Fig. 15a, we used the energy minimized hinge structure mentioned above, then added lysine to the N-terminal side of the hinge peptide dimer and used it to align K237 of the heavy chain of PDB 1TJG (https://www.ncbi.nlm.nih.gov/protein/1TJG_H) and join the two Fabs as shown. In this model, the upper hinges form about 72-degree angle. Models in Fig. 6a provides a 'relaxed' state with the same orientations of hinges and Fabs, while Supplementary Fig. 15a provides a model where IgG1 is nearly fully

open, yet the IgG3 is not stretched as much. Given the near 180° angle between the two Fabs in IgG1 (Supplementary Fig. 15a), 155 Å is likely an overestimate for the span of IgG1, so that the difference between IgG1 and IgG3 in their capacity to crosslink between two spikes on the virion would be an underestimate, which will be even greater when the angular range of the Fab domains are taken into account. More detailed analysis of the spanning distance and angular range will be carried out in a future study.

## Building models of a virion

The virion surface was approximated as a sphere of radius $R = 55$ nm[9]. To place a spike at a randomly selected location on the surface of the sphere, three Gaussian-distributed and statistically independent random numbers $x_0, y_0, z_0$ with zero average and with unit variance were generated. Their joint probability distribution $P(x_0, y_0, z_0)$ $\sim \exp\{(x_0^2 + y_0^2 + z_0^2)/2\} = \exp(r_0^2/2)$, is independent of the direction of the vector $\mathbf{r_0} = (x_0, y_0, z_0)$, where $r_0 = |\mathbf{r_0}|$. Thus, a spike was placed at position $(x_0, y_0, z_0) \times R/r_0$. The radius and height of the spike was 5.25 nm and 13.7 nm, respectively. When adding a new spike, if its center-to-center distance to the closest one among previously added spikes was less than the diameter of the spike (10.5 nm), it means the spike to be added contacts a neighboring spike. This can be seen from an analogous observation that two overlapping circles with the same diameter have center-to-center distance less than their diameter. In this case, a new set of random numbers $(x_0, y_0, z_0)$ were generated to find a new position on the sphere. This process continued until the new position did not cause any overlap between spikes. The surface-to-surface distance $s$ between two spikes was measured using the angle $\theta$ between them on the sphere, subtracted by the diameter $d$ of the spike, as $s = R\theta - d$.

## Statistics

For flow-cytometry binding assays, results acquired by two or three independent experiments are presented as mean ± SEM (standard error of the mean), unless stated otherwise. No statistical methods were used to predetermine sample sizes. No data were excluded from analysis. The exact sample size or number of experiments performed is mentioned in figure legends. Statistical significance of data was determined by two-way analysis of variance (ANOVA) followed by Šídák's multiple comparisons test. The p-value indicates highly significant ****$p < 0.0001$ while $p > 0.05$ is considered not significant. Graphical representations and data analysis were performed using Graphpad prism 8 software. For all mAb pseudovirus neutralization, the $IC_{50}$ or concentration of mAb needed to obtain 50% neutralization against a given pseudovirus was calculated from the linear regression of the linear part of the neutralization curve. For ELISA, $EC_{50}$ of antibody was calculated from linear regression of the linear part of antibody binding curve. For measuring average number of crosslinkable spike pairs (Fig. 6c, d and Supplementary Fig. 15c, d), due to the large number of models generated (100,000 for $N = 14$ and 20,000 for $N = 73$), the values vary by at most 3% when average and standard deviation are measured individually for 10 subsets of data ($p < 10^{-16}$).

## Reporting summary

Further information on research design is available in the Nature Portfolio Reporting Summary linked to this article.

## Data availability

Atomic coordinates and structure factors for the reported crystal structures have been deposited in the Protein Data Bank under accession numbers: 8FWF, 8FYM, 8FXJ and 8FZ2. All data generated or analyzed during this study are included in this article and its supplementary information file. Source data are provided with this paper.

## Code availability

C++ source code and Python programs to produce models in Fig. 6b and distributions in Fig. 6c, d and Supplementary Fig. 14 are available for download from https://github.com/hwm2746/virion-spike-model. https://doi.org/10.5281/zenodo.8287823. Models and data for Supplemental Fig. 15b–d can be obtained by modifying input parameters in this code, as explained in README.md file in the GitHub repository.

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

## Acknowledgements

We thank Jia-huai Wang for his insightful comments. We thank Gilad Ofek for initial crystallization trials. We thank Peter Kwong for kindly providing expression plasmids encoding 10E8 IgG1 heavy and light chain, 4E10 IgG1 heavy and light chain, and VRC01 IgG1 heavy and light chain. The following reagents were obtained through the NIH HIV Reagent Program, Division of AIDS, NIAID, NIH: plasmids encoding HIV-1 Envs derived from 96ZM651.8 (ARP-8665), JR-FL (ARP-4599), VC20013_100506_c19 (ARP-13336), and human soluble CD4 recombinant protein (domains D1-D4) (ARP-4615). This work was supported by AI145509 to M.K.; AI126901 to E.L.R. and T.W. The use of SBC 19-ID at Argonne National Laboratory was supported by DOE contract no. DE-AC02-06CH11357.

## Author contributions

Conceptualization: M.K., E.L.R.; Antigen-specific B-cell screening: L.D.; Molecular cloning of various antibodies: L.D., Y.W., J.C. R.K.; Expression and purification of proteins: J.C., Y.W., Y.K., R.K., Antibody function experiments: J.C., Y.W., Y.K., R.K.; Neutralization assay: H.R., M.S.S.; Crystallization: X.L., K.T.; X-ray crystallography data collection, processing and model building: K.T.; Building models of IgG1 and IgG3, and virions: W.H.; Writing or editing of manuscript: M.K., K.T., T.W., W.H., E.L.R.

## Competing interests

The authors declare no competing interests.
