## [Peer Review File · Nature Communications]

Inadequate structural constraint on Fab approach rather than paratope elicitation limits HIV-1 MPER vaccine utilityREVIEWER COMMENTS

Reviewer #1 (Remarks to the Author):

- 1) The authors provide experimental results that seem to suggest that MPER-liposome vaccines elicit antibodies with sub-optimal angles of attack for accessing epitopes in vivo, presumably because these vaccine constructs lack the steric reality of the rest of the peplomer; this seems like a useful scientific insight, though I'll leave it to experimentalists or immunologists to chime in on whether it is a novel finding
- 2) The findings related to IgG3 vs. IgG1 for germline antibodies are a bit hard for me to judge--perhaps for those of us who are not immunologists, it would be helpful to clarify how long (wallclock/calendar time) the germline antibodies would be prevalent relative to mature antibodies in an HIV-1 infected individual? Combined with the described low abundance of IgG3, I wasn't entirely sure that it was decisively clear what we could conclude related to the overall clinical importance of IgG3 hinge flexibility?
- 3) The Env/spike protein is known to be $\sim 1/2$ glycans by mass, but I don't see much analysis/discussion about the potential complication of the glycan shield in the compared antibody/Fab angles of attack. What is going on with glycans near the various potential attack angles?
- 4) The work related to Figure 6 is where I can primarily comment from the computational/modeling side:
 - a) Obviously, no physics-based simulations were performed, and as indicated in the manuscript, the Fab domains are placed "arbitrarily" to provide an approximate estimate of their spans, so I don't think we can use any of this "modeling" to substitute for any weaknesses in the experimental work. I'm not suggesting that are issues with the experimental work, nor that we should remove the modeling Figure, just that it is mostly a schematic to aid with visualization rather than anything physically realistic.
 - b) If we exclude SIV, the relevance of which is less clear to me here, the HIV-1 bivalent binding options are only moderately enhanced (~ 1.5 extra cross-link) for IgG3 vs. IgG1; is there compelling literature evidence that the antibodies that bind SIV have extremely similar structural properties to the human antibodies that bind HIV-1? couldn't the increased peplomer density in SIV lead to reduced selective pressure for inter-Fab flexibility? there seem to be a lot of variables here, and their consideration in the discussion may just complicate matters, so I'm wondering if the SIV part of the "results" makes sense to include?
 - c) MD simulations involving the clustered "entry claw" of peplomers described in the discussion and the germ-line versions of the antibodies may be interesting, though this would be a substantive undertaking even for a group specialized in such simulations.
 - d) I don't see the source code for the virion model generation/analysis provided/referenced in the main manuscript--shouldn't that be made available on i.e., GitHub for inspection? The method itself sounds fairly standard/simple--a Gaussian distribution is a sensible choice for random placement on the sphere.
 - e) One other comment regarding the virion models--do the radii of these spike models include or not include the large glycan shield? It might also be sensible to cite the source(s) for the radius and heights of the spikes in the methods section? I'm perhaps not clear

on why the height is even relevant if the focus is on MPER binding proximal to the surface?

Overall, the manuscript appears to tell an interesting story about MPER-liposome vaccine elicitation, and certainly seems to involve a large volume of structural biology and immunology experiments in support of it. The merit of the experimental side should be judged by appropriate experts though. As for the computational side, since this appears to be primarily a visual aide in Figure 6, I don't have any major objections to it, but I would say it shouldn't be considered to make up for any concerns experimentalists might have, since no physics-based simulation was done beyond checking some very basic geometric constraints.

Reviewer #2 (Remarks to the Author):

Tan et al studied the geometric interplay between vaccine induced Abs and HIV MPER. They show that certain vaccines have failed to elicit Abs that can access MPER. As a solution they suggest attempting to generate IgG3 which are longer and apparently more flexible. There are a number of interesting ideas and appears to be a serious amount of work to characterize the spatial aspects of these Abs and how they bind. However, I had several major concerns about the applicability/comprehensibility of the current manuscript:

Starting with the title: "Lack of Fab approach constraint and not paratope elicitation limits HIV-1 MPER vaccine utility" -- lack of constraint is a difficult double negative but more importantly this doesn't seem to really evoke the conclusions of the paper: 1) a certain MPER vaccine mostly elicited Abs that could not bind due to geometry. 2) IgG3 Abs might overcome this. Paratope elicitation was not mentioned much in the manuscript and to create a convincing narrative, the authors need to demonstrate that until this work there was a big misconception about why this vaccine did not work.

There were also several statements in the introduction that seemed unusually strong. For instance they state things like: "preventive vaccines essential to terminate the AIDS pandemic (1)" -- how do you know this will be essential? is this reference saying this? I think prevention is crucial (ie <https://www.pnas.org/doi/10.1073/pnas.1703236114>) but wouldn't universal PrEP accomplish this too? Authors also state: "an effective vaccine must be precisely fashioned to elicit broadly neutralizing antibodies (bnAbs) that bind to conserved epitopes on the HIV-1 trimeric gp160 envelope (Env)", but how do they know this is the the only way to make an HIV vaccine? They state "the sparsely arrayed gp160 spikes represent the only target for protective humoral immunity" -- a reference for this is needed and has humoral immunity even been shown to be protective in humans? They state "MPER-specific bnAbs do arise after several years of infection (10)" this reference is rather specific, and it seems to show these Abs can arise in certain cases, not that they are guaranteed to.

On modeling, the methodology could be better explained/written (see minor notes). Nevertheless, a more complete analysis is required if the goal is to provide "broad implications for vaccine design against infectious pathogens." For starters, it should explore more of the different variables that are mentioned in this paper: number of spikes, size of spikes, size of Abs -- As an aside the figure legend mentions the 15nm size is an overestimate, so a sensitivity analysis on this parameter is probably useful. Overall, testing these additional variables does not seem very difficult when a sphere is assumed and the spikes are dotted around it. More importantly, it wasn't clear to me if the main concept from the experimental studies (the flexibility of different IgGs to reach the location) was not modeled explicitly, such that this could be optimized and/or balanced against the size advantage for the number of cross-links. Finally, it would really help to output a more comprehensive result which is the absolute probability of neutralization, rather than the probability of cross-linking.

With respect to the modeling results, these were somewhat difficult for me to interpret. The text asserts that crosslinking is more important for low-target situations, however, the data from the simulations shows overlapping distributions of pairs for HIV (low N) and distinct distributions for SHIV

(high N). This makes it look like the advantage of IgG3 vs IgG1 increases with targets. Maybe the point is that once you get enough cross-linking, you don't need more?, but these simulations do not address this diminishing returns concept.

Finally, there seems to be an attempt in the discussion and at different places in the manuscript to explain how the present findings are relevant for anti-viral vaccines more broadly, but this was scattered and needs to be much more specific.

Minor:

intro:

given that the -- extra "that"?

While... trimers -- need refs

paradox -- not sure what the paradox is, usually needs to be something self-contradictory

disfavor -- is this a verb?

HIV-1 infected patients -- often now phrased "people living with HIV" but double check latest standards

Methods:

"it means the spike to be added contacts a neighboring spike" -- something is a little hard to parse here

"new position does not cause any overlap between spikes" -- tense issue

"surface-to-surface distance" -- maybe write this out as an equation since it requires distance on sphere

Reviewer #3 (Remarks to the Author):

Tan et al., "Lack of Fab approach constraint and not paratope elicitation limits HIV-1 MPER vaccine utility" presents reasonable, well-supported explanations for a number of conundrums around the neutralization mechanism of anti-HIV MPER bnAbs. The scope of the work is impressive, and experiments are well described, rigorously performed, and properly interpreted across methodologies. While the presentation is expansive, reflecting the breadth of the results, the work is overall cohesive and just a "good read" as-is, so I'm not recommending any cuts. Indeed, with only a very few, mostly optional, comments, the manuscript should be accepted for publication. The selection of venue also seems completely appropriate, in terms of both content and impact.

Minor comments:

Fig. 2 legend: "Residues L30L and Y31L on the protruding CDR1L of Fab235 are drawn in stick representation to show their potential interactions with the membrane as is MPER": since the residues in the most exposed section of this loop seem to be YSSNQK, which don't strike me as particularly "membrane interactive", perhaps reconsider this statement.

Is the Fab235+MPER-C structure, with $z = 4$, really P1? Just a quick comment in the crystallography section confirming this might be appropriate.

I'd strongly recommend one more read-through of the manuscript, as there are several typos, awkward wording, and mis-formatting in this current draft.

Otherwise, nice job.

Dear Reviewers,

Our responses to the queries in black font are given in blue font below each, point by point. Main text changes in the revised manuscript are called out in red font.

Reviewer #1 (Remarks to the Author):

1) The authors provide experimental results that seem to suggest that MPER-liposome vaccines elicit antibodies with sub-optimal angles of attack for accessing epitopes *in vivo*, presumably because these vaccine constructs lack the steric reality of the rest of the peplomer; this seems like a useful scientific insight, though I'll leave it to experimentalists or immunologists to chime in on whether it is a novel finding.

We appreciate the reviewer's candor regarding his (her) expertise as well as limitations.

2) The findings related to IgG3 vs. IgG1 for germline antibodies are a bit hard for me to judge--perhaps for those of us who are not immunologists, it would be helpful to clarify how long (wallclock/calendar time) the germline antibodies would be prevalent relative to mature antibodies in an HIV-1 infected individual? Combined with the described low abundance of IgG3, I wasn't entirely sure that it was decisively clear what we could conclude related to the overall clinical importance of IgG3 hinge flexibility?

The comments on germline antibodies are meant to refer to antigen-specific B cells arising early in an ongoing immune response, not the antibody pharmacologic half-life *per se*. Protective neutralizing antibodies are produced by long-lived bone marrow plasma cells and plasmablasts that differentiate from activated memory B cells. Affinity maturation of neutralizing antibodies is achieved via iterative rounds of somatic hypermutation (SHM) and B cell selection in transient germinal center (GC) reactions in lymph nodes and spleen, wherein the affinity of a B cell receptor (BCR) against an antigen increases. As GC B cell selection is dependent on and positively correlated with the affinity of the BCR for the antigen, relatively high affinity B cells preferentially survive and are chosen for further affinity maturation. Conversely, B cells with low affinity undergo apoptosis. In this competitive environment, advantage contributed by BCR avidity would benefit the B cell selection from early on in the GC reaction, particularly when the SHM rate of the BCR is low (and hence affinity is low). In that regard, given that HIV spike density is sparse and MPER accessibility is restricted, the IgG3 subclass with improved capacity to foster MPER accessibility and inter-spike crosslinking would provide an advantage from the early stage of the immune response for those B cells to be positively selected for further affinity maturation over IgG1, eventually developing into neutralizing antibodies. That is the likely reason for the abundance of MPER-specific IgG3 bnAbs arising during infection of individuals with HIV-1. As our results suggested, biological functions of IgG3 such as their crosslinking efficiency/avidity and MPER accessibility are superior to IgG1. In addition, Fc-mediated effector function of IgG3 has been shown to be better than that of IgG1. Those functions would contribute to enhanced protection compared to IgG1 when an infected individual is exposed to heterologous viral strains. Therefore, even though the overall prevalence of IgG3 is low relative to other immunoglobulin subtypes at steady state, a vaccine regimen that can recruit and maintain higher IgG3 isotype switched B cells during GC reactions would provide a better chance to induce high-affinity neutralizing antibodies against the MPER.

3) The Env/spike protein is known to be ~1/2 glycans by mass, but I don't see much analysis/discussion about the potential complication of the glycan shield in the compared antibody/Fab angles of attack. What is going on with glycans near the various potential attack angles?

This is an excellent point. We now provide new Supplementary Figure 12 to show the position of all glycans on the HIV-1 spike, including those near the MPER, as well as functional data demonstrating that removal of such glycans has no effect on 10E8 bnAb binding. We appreciate the reviewer's suggestion.

4) The work related to Figure 6 is where I can primarily comment from the computational/modeling side:
a) Obviously, no physics-based simulations were performed, and as indicated in the manuscript, the Fab domains are placed "arbitrarily" to provide an approximate estimate of their spans, so I don't think we can use any of this "modeling" to substitute for any weaknesses in the experimental work. I'm not suggesting that are issues with the experimental work, nor that we should remove the modeling Figure, just that it is mostly a schematic to aid with visualization rather than anything physically realistic.

As with many diffusion-limited collision process between molecules in fluid, it is expected that initial encounter between Fab domains and the virion is random. If one Fab domain happens to bind tightly to the MPER of a spike, the other Fab of the antibody will move on the virion surface and may bind to a second MPER. For this process, the span of the two Fab domains and spacing between spikes are two factors that sterically determine whether inter-spike crosslinking is possible. The binding affinities of Fab domains to MPER being similar between IgG1 and IgG3, the steric factors should play a major differentiating role in their biological functions.

Had there not been the quantitative analysis examining the spike distance distribution and the difference between spans of IgG1 versus IgG3, it would not have been possible to conclude whether the steric aspects are important for the experimentally observed differences. This point is made evident by our comparative analysis of the SIV virion, where difference between spans of the two antibodies is less critical compared to HIV-1. Thus, our model is not only a computationally-derived visual aid, but supports an explanation for the experimental wet-lab results on differences in affinity and neutralization potency between germline bnAbs of IgG1 vs IgG3 subtypes (Fig. 5) and provides additional insight into the interplay between viral spike density and anti-viral antibody function.

b) If we exclude SIV, the relevance of which is less clear to me here, the HIV-1 bivalent binding options are only moderately enhanced (~1.5 extra cross-link) for IgG3 vs. IgG1; is there compelling literature evidence that the antibodies that bind SIV have extremely similar structural properties to the human antibodies that bind HIV-1? couldn't the increased peplomer density in SIV lead to reduced selective pressure for inter-Fab flexibility? there seem to be a lot of variables here, and their consideration` in the discussion may just complicate matters, so I'm wondering if the SIV part of the "results" makes sense to include?

Non-human primate (NHP) retroviruses from chimpanzees evolved into HIV-1 during interspecies transmission. Unsurprisingly, the related monkey SIVmac Env trimer shows overall conserved secondary and tertiary structure with HIV-1 Env trimer and only minor deviations in variable loops and the glycan shield. The SIV Env trimer functions, such as CD4 receptor and co-receptor binding as well as virus fusion, are also well conserved. It has been reported that neutralization mechanisms of bnAbs against SIV Env trimer (directed to gp120 subunit) are also similar to that of bnAbs against HIV-1 Env trimer. We provide these relevant references.

1) Molecular insights into antibody-mediated protection against the prototypic simian immunodeficiency virus *Nat Commun.* 2022 Sep 6;13(1):5236 ; PMID: 36068229

2) Cryo-EM structures of prefusion SIV envelope trimer. *Nat Struct Mol Biol.* 2022 Nov;29(11):1080-1091; PMID: 36344847

3) A Germline-Targeting Chimpanzee SIV Envelope Glycoprotein Elicits a New Class of V2-Apex Directed Cross-Neutralizing Antibodies. *mBio.* 2023 Feb 28;14(1): e0337022. ; PMID: 36629414

While anti-SIV MPER specific bnAbs have not been reported, we anticipate that the neutralization mechanism of such antibodies would be the same. We took the opportunity to investigate the effect of spike density on the antibody cross-linking efficiency modulated by the hinge of different IgG subtypes, since the SIVmac Env variant has a high density of spikes relative to HIV-1 and in the context of a

structurally related virus of similar size. Explicitly, SIV with N=73 spikes was selected in comparison with HIV-1 (N=14 spikes) as the only variable factor since the size of virus, size/structure of Env trimer and biological functions of Env trimer between SIV and HIV-1 are very similar. As the reviewer pointed out, the increased peplomer density in SIV leads to reduced selective pressure for inter-Fab flexibility. By directly comparing the 73 vs. 14 spike numbers, we think that the benefit of IgG3 against HIV-1, and by extension, other viruses with low density of spikes with similar size is clear, as mentioned in our response 2 and 4a above. Predicated on additional comments made by reviewer 2, we have now provided in the main Fig 6 (new) and Supplementary Figure 15 (revised from the old Fig 6) data using two intra-immunoglobulin Fab-Fab arm distances for the IgG1 vs. IgG3 anti-MPER specificities [(99Å vs. 166Å) and (155Å vs. 192Å), respectively] using the geometries shown therein. The former pair is a more realistic distance for the IgG molecule, as emphasized by reviewer 2 below, but the more relaxed hinge with larger distance used in the earlier Fig.6 version has the same trend as shown in the Supplementary Figure 15.

Regarding moderately enhanced crosslinking of inter-spikes by IgG3 compared to IgG1, the 2-fold increase is likely significant, especially when Ab affinity is suboptimal or reduced as observed in our experimental results (Fig. 5). Given low density of HIV-1 spikes on the virion surface, a small number of trimers required for virus fusion and sequence variations between diverse HIV-1 strains/isolates, the reduced bnAb binding affinity for heterologous strains can be compensated by enhanced crosslinking of inter-spikes by IgG3. The increment in crosslinked spikes may be sufficient to keep the sparse number of spikes from mediating virus fusion. Furthermore, our earlier Fig. 6 (now updated as Supplementary Figure 15) compares IgG1 in nearly a 180-degree stretch of Fab-Fab arms with IgG3 that is less stretched. In the new Fig. 6 comparing the subtypes in a more 'relaxed' state, the difference increases to 2.4. This comparison is based on steric considerations only, but as the new Fig 6A and discussion indicates, the orientational flexibility of the Fab domains will be more limited in IgG1, further reducing its capacity to crosslink peplomers. This point is made clearer in the revised results and discussion section.

c) MD simulations involving the clustered "entry claw" of peplomers described in the discussion and the germ-line versions of the antibodies may be interesting, though this would be a substantive undertaking even for a group specialized in such simulations.

We agree that simulation describing the formation and action of the entry claw would be highly valuable. As the reviewer correctly points out, this will be a significant undertaking, requiring experimental data about their structure and dynamics that are beyond the scope of the current work. Given the size of the system and time scales involved, instead of molecular dynamics (MD) simulation, coarse-grained modeling and simulation would be more appropriate. This would be subject of a future study.

d) I don't see the source code for the virion model generation/analysis provided/referenced in the main manuscript--shouldn't that be made available on i.e., GitHub for inspection? The method itself sounds fairly standard/simple--a Gaussian distribution is a sensible choice for random placement on the sphere.

Given that the method uses a fairly standard way of generating random locations on the virion surface and the description in Methods which should be sufficient to allow an individual with a moderate level of programming to reproduce the result, we did not consider our codes to be extensive enough to deposit in GitHub. However, in view of the reviewer's suggestion, we have deposited it to be available at <https://github.com/hwm2746/virion-spike-model>

e) One other comment regarding the virion models--do the radii of these spike models include or not include the large glycan shield? It might also be sensible to cite the source(s) for the radius and heights of the spikes in the methods section? I'm perhaps not clear on why the height is even relevant if the focus is on MPER binding proximal to the surface?

The MPER region is highly conserved amongst viral clades and strains and is very close to the virion surface, being accessible without any glycan directly covering it as discussed above in response 3. This is why the MPER-directed antibodies hold great potential for mediating broad neutralization. The 5.25-nm radius is based on the known structure of the gp160 trimer (spike protein), which is independent of the extent of the glycan shield in other parts of the protein. Using a radius slightly different from 5.25 nm is not going to affect the main conclusion from our modeling analysis about the steric advantage of IgG3 over IgG1 for inter-spike crosslinking. We used the height of the trimer to just provide a visual sense of its size in our model (Fig. 6B). However, our measurement was based on the surface-to-surface distance, as explained in Methods.

Overall, the manuscript appears to tell an interesting story about MPER-liposome vaccine elicitation, and certainly seems to involve a large volume of structural biology and immunology experiments in support of it. The merit of the experimental side should be judged by appropriate experts though. As for the computational side, since this appears to be primarily a visual aide in Figure 6, I don't have any major objections to it, but I would say it shouldn't be considered to make up for any concerns experimentalists might have, since no physics-based simulation was done beyond checking some very basic geometric constraints.

As noted in the response to point 4a above, our model is not simply a visual guide, but rather helps to elucidate the difference between IgG1 versus IgG3 in their neutralizing capacity, specifically on the HIV virion surface with a sparse number of spike proteins. Furthermore, our functional data are statistically robust and stand on their own merit.

Reviewer #2 (Remarks to the Author):

1) Starting with the title: "Lack of Fab approach constraint and not paratope elicitation limits HIV-1 MPER vaccine utility" -- lack of constraint is a difficult double negative but more importantly this doesn't seem to really evoke the conclusions of the paper: 1) a certain MPER vaccine mostly elicited Abs that could not bind due to geometry. 2) IgG3 Abs might overcome this.

We thank the reviewer for his/her care in review of our manuscript. We have updated the title to: "Inadequate structural constraint on Fab approach rather than paratope elicitation limits HIV-1 MPER vaccine utility"

While the IgG3 subtype of vaccine-elicited antibodies enhanced MPER binding compared to that of IgG1, the improvement was insufficient to generate effective neutralization. The role of the flexible hinge of IgG3 can't overcome binding resulting from an inadequate Fab approach angle. Thus, while we agree the IgG3 data is important, it cannot be added to the title. Elicitation of antibodies with the correct approach angle should be a priority in immunogen design. When this criterion is met, however, induction of IgG3 over IgG1 subtype would benefit biological functions especially when serum antibodies are exposed to heterologous viruses for the reasons enumerated and mentioned in response to question 1 from reviewer 1.

2) Paratope elicitation was not mentioned much in the manuscript and to create a convincing narrative, the authors need to demonstrate that until this work there was a big misconception about why this vaccine did not work.

We thank the reviewer for the suggestion and have modified the text in the introduction as indicated in red text font on page 3-4, line 80-83.

3) There were also several statements in the introduction that seemed unusually strong. For instance they state things like: "preventive vaccines essential to terminate the AIDS pandemic (1)" -- how do you know this will be essential? is this reference saying this? I think prevention is crucial (ie <https://www.pnas.org/doi/10.1073/pnas.1703236114>) but wouldn't universal PrEP accomplish this too?

We have amended this sentence to state “preventative vaccines that may play a critical role in controlling the HIV epidemic”. A revised reference (1) has further been added which better addresses the hurdles of sequence diversity and the important role that vaccines may play in prevention of HIV-1 infection. While the authors agree with the reviewer that PrEP has become a major component of HIV-1 prevention efforts, there are also limitations associated with treatment-as-prevention strategies, in particular global access, and compliance. A comparison between these approaches is beyond the scope of the current manuscript.

4) Authors also state: "an effective vaccine must be precisely fashioned to elicit broadly neutralizing antibodies (bnAbs) that bind to conserved epitopes on the HIV-1 trimeric gp160 envelope (Env)", but how do they know this is the only way to make an HIV vaccine? Has humoral immunity even been shown to be protective in humans?

We have amended this sentence to highlight the recent failures of other HIV-1 vaccine design efforts that elicited CD8+ T cell responses and non-neutralizing antibodies to demonstrate protection in clinical efficacy trials (red font on page 3, line 52-54). While these effector arms of the immune system likely contribute to protective immunity against HIV-1, it is widely believed that for a vaccine to demonstrate efficacy in reducing HIV-1 transmission it must have the capacity to elicit broad and potent neutralizing antibodies. Antibody passive immunization studies in NHP and humans support this hypothesis. With regard to the important role of humoral immunity in vaccine-mediated protection in humans, we point the reviewer to the following article: Plotkin, S.A. (2020). Updates on immunologic correlates of vaccine-induced protection. *Vaccine* 38, 2250-2257.

5) They state "the sparsely arrayed gp160 spikes represent the only target for protective humoral immunity" -- a reference for this is needed

We have added a reference (5) upon the reviewer’s suggestion.

6) They state "MPER-specific bnAbs do arise after several years of infection (10)" this reference is rather specific, and it seems to show these Abs can arise in certain cases, not that they are guaranteed to.

We have amended the sentence: MPER-specific bnAbs can arise after several years in a subset of infected individuals (line 69-70 in page 3).

7) On modeling, the methodology could be better explained/written (see minor notes). Nevertheless, a more complete analysis is required if the goal is to provide "broad implications for vaccine design against infectious pathogens." For starters, it should explore more of the different variables that are mentioned in this paper: number of spikes, size of spikes, size of Abs -- As an aside the figure legend mentions the 15nm size is an overestimate, so a sensitivity analysis on this parameter is probably useful. Overall, testing these additional variables does not seem very difficult when a sphere is assumed and the spikes are dotted around it. More importantly, it wasn't clear to me if the main concept from the experimental studies (the flexibility of different IgGs to reach the location) was not modeled explicitly, such that this could be optimized and/or balanced against the size advantage for the number of cross-links. Finally, it would really help to output a more comprehensive result which is the absolute probability of neutralization, rather than the probability of cross-linking.

As reviewer 2 pointed out, we used an over-estimate for the span of IgG1 while a more reasonable span of IgG3 was used, to elucidate the higher capacity of the latter in crosslinking between spikes even in such a 'disadvantaged' situation. This created confusion for which we apologize. In response to this comment (also response to reviewer 1, point 4) and to provide greater clarity, we made two models explicitly given in the revised manuscript, one placed in Fig 6 and a second in Supplementary Figure 15. The former uses the same angle between the two arms of the hinge and orientation of Fab domains of the Ab subtypes. This leads to 99 Å as the Fab-Fab span of IgG1 that is more realistic compared to the 155-Å span of IgG1 noted previously. The span of IgG3 is likewise reduced in Fig. 6A compared to that in

Supplementary Figure 15. The trend for HIV-1 vs. SIV spike crosslinking is clear, despite use of these disparate dimensions as described further in response 4b to referee 1.

Regarding absolute probability of neutralization rather than the probability of cross-linking, first, we thank the reviewer for the comment. We were motivated to perform modeling analysis based on our experimental result of differences in neutralizing potency between IgG1 and IgG3 subtypes. Because the paratope site is the same and Fc-mediated effector function is excluded in our neutralization assay (i.e. no Fc receptor on target cells), the question as to whether the modeling analysis could provide information on the number of inter-spike crosslinks mediated by different subtype hinges was straightforward in principle. That data would provide insight into the neutralization potency modulated by antibody subtype and spike density. On the other hand, calculating absolute probability of neutralization is significantly more complex. Because HIV-1 virions display non-functional trimer species, it is not possible to distinguish between the total number of virion trimers and those that are functional at the time of virus infection. In addition, variation of trimer numbers across virions and between strains impacts virus infectivity. The number of trimers required for virus infection and the amount of bnAb required to inhibit the process may vary accordingly, and the analysis may foster predictions inconsistent with observed functional data in *in vivo*. Notwithstanding, more comprehensive result including absolute probability of neutralization would be ideal, as suggested by the reviewer, but requires inclusion of many variables in such a model, well beyond the extensive structural and functional data already amassed here.

8) With respect to the modeling results, these were somewhat difficult for me to interpret. The text asserts that crosslinking is more important for low-target situations, however, the data from the simulations shows overlapping distributions of pairs for HIV (low N) and distinct distributions for SHIV (high N). This makes it look like the advantage of IgG3 vs IgG1 increases with targets. Maybe the point is that once you get enough cross-linking, you don't need more?, but these simulations do not address this diminishing returns concept.

The significantly overlapping distributions were due to the fact that we used an over-estimate for the span of IgG1, as mentioned above. New measurement with a more relaxed span of IgG1 (dashed line in Fig 6C,D) shows a greater separation in the average number of crosslinkable pairs. As explained in our response 4b to reviewer 1, the difference between the two antibodies in crosslinking capacity will increase further when differences in the orientational flexibility of the Fab domains is accounted for, being more limited for IgG1. About the separation of peaks in Fig. 6D and Supplementary Figure 15D, it is because there are many more potentially crosslinkable protomers in SIV compared to HIV. The percentage of protomers that can be crosslinked can be estimated based on Supplementary Figure 14 showing distributions of the nearest-neighbor distance pairs. Values corresponding to distances considered in Fig. 6 and Supplementary Figure 15 are:

Distance	Cumulative distribution (red line in Supplementary Figure 14)
~10nm	23% (N=14), 90% (N=73) <- IgG1
~17nm	48% (N=14), 99.5% (N=73) <- IgG3

For N=14 (HIV), 23% and 48% of the protomers have nearest neighbors with distances within 10 and 17nm, respectively. So even IgG3 can crosslink slightly less than 50% of the protomers. The percentages become 90% (IgG1) and near-100% (IgG3) when N=73. In both cases, the majority of protomers can be crosslinked. Thus, separation of peaks in Fig. 6D and Supplementary Figure 15D does not translate to differences in neutralization capacities of IgG1 and IgG3 against SIV. This explanation regarding Fig. 6 is now mentioned in the main text on page 11, line 328-332.

9) Finally, there seems to be an attempt in the discussion and at different places in the manuscript to explain how the present findings are relevant for anti-viral vaccines more broadly, but this was scattered and needs to be much more specific.

We thank the reviewer's suggestion. We modified text accordingly in the discussion section on page 12, line 357-370 and line 388.

10) disfavor -- is this a verb?

We modified the sentence on page 4, line 84 shown in font change.

11) Methods: "it means the spike to be added contacts a neighboring spike" -- something is a little hard to parse here

This simply reflects the fact that when the center-to-center distance between two circles is less than the diameter of the circle, the two circles overlap. The text has been updated for clarity in the methods section "**Building models of a virion**" on page 20, line 609-610

12) "new position does not cause any overlap between spikes" -- tense issue

This has been corrected. Thanks.

13) "surface-to-surface distance" -- maybe write this out as an equation since it requires distance on sphere

Equation describing this has been added in the last sentence of the section "**Building models of a virion**" on page 20, line 613-614.

Reviewer #3 (Remarks to the Author):

1) Regarding comment on Fig. 2 legend: "Residues L30L and Y31L on the protruding CDR1L of Fab235 are drawn in stick representation to show their potential interactions with the membrane as is MPER": since the residues in the most exposed section of this loop seem to be YSSNQK, which don't strike me as particularly "membrane interactive", perhaps reconsider this statement.

We appreciate the reviewer's overall positive comments. We have modified the sentence in Fig. 2 legend as follows: For clarity, residues L30 and Y31 on one side of the protruding CDR1L of Fab235 are marked in stick representation to denote potential interactions of the CDR1L loop with the membrane. This loop also includes residues, S₃₂SNQK₃₆ that may undergo conformational change upon Fab binding to the MPER. For example, K36 may also participate in interactions with the phosphate head groups of membrane lipids.

2) Is the Fab235+MPER-C structure, with $z = 4$, really P1? Just a quick comment in the crystallography section confirming this might be appropriate.

We thank the reviewer for the suggested clarification, especially for general readers. We have added one note (Note #2) in Supplementary Table 1 "Data collection and refinement statistics by $Z=4$ " that states that the four complexes in one asymmetrical unit are distinguishable and, hence, there is no extra symmetry for a space group of higher symmetry. This possibility was also excluded during data processing.

Sincerely,

Mikyung Kim
Ellis Reinherz

REVIEWERS' COMMENTS

Reviewer #1 (Remarks to the Author):

I'm mostly satisfied that my comments were addressed, and the authors did some extra experimental work to address my concerns about the glycan shield as well as making the source code used for the model work available to the public on GitHub.

I don't think I have any additional requests for changes, but I'll just note that:

1) It may be sensible for one of the other two reviewers to double check the rebuttal to point #2 (the long response about B cells, lymph nodes, and spleen, is a bit outside my remit).

2) Use caution in putting too much value in the geometric analyses in Figure 6, given lack of physics-based simulations. But I'm happy with the additional effort added in there, plus I think this journal will make the back and forth discussion available for public inspection, so folks can decide on their own.

Tyler Reddy, Los Alamos National Laboratory

Reviewer #2 (Remarks to the Author):

I appreciate the in depth response to mine and the other reviewers comments. Although I am not able to comment on the quality of the experiments/data, the structural work appears important and the results interesting. Unfortunately, the authors basically declined to modify/enhance the modeling despite comments from myself and reviewer 1 that the model is relatively simple as to not very convincingly describe the phenomenon in detail. As is, we do not gain much quantitative insight other than longer is better for crosslinking. I also do not think the authors took very seriously the comments that the paper is hard to read for non-structural biologists familiar with HIV vaccines.

Minor, but there are still several problematic statements in the introduction:

"prevent virus entry into the host" -- is this really what bnAbs do? don't they neutralize/clear virus after it enters the mucosal tissue? SHIV experiments by Liu et al. and Hessel et al. indicate systemic virus spread before clearance

"HIV-1-positive individuals" - I believe people living with HIV is favored

"outstanding neutralization breadth" - would be important to provide some quantitative numbers here, e.g. in the Huang paper you cite they say 10^8 neutralized 98% of tested viruses, but there must be more up to date information on this

If the authors goal is to influence vaccine research broadly, I do think it would be really important to try to simplify the abstract. The sentences at the end of the introduction summarizing the study are clearer to me than the current abstract. I also attempted to write a version that the authors could borrow from, with the hope it helps:

Broadly neutralizing antibodies (bnAbs) against HIV-1 target conserved epitopes to block viral replication. Here, using structural analyses, we explain why a vaccine targeting the membrane proximal external region (MPER) of HIV which elicited antibodies with human bnAb-like paratopes was unable to bind HIV. Unlike in natural infection, vaccination lacks a structure-based selection mechanism such that resulting Abs cannot physically access the MPER crawlspace on the virion surface. By studying naturally induced Abs, we show flexibility of the human IgG3 hinge overcomes this structural inaccessibility and additionally facilitates HIV spike protein crosslinking. Our results suggest inducing IgG3 subtype class-switched B cells as a future target for HIV bnAb induction. Moreover, they broadly illustrate how immunogen design must consider structural matching between immunogen targets and pathogen features.

Reviewer #3 (Remarks to the Author):

The authors have adequately addressed all outstanding concerns raised during review so the improved/revised manuscript should be accepted for publication.

Reviewer #4 (Remarks to the Author):

The authors have adequately responded to the first three reviewers.

I have only minor comments.

1. the usual reason for lack of MPER antibodies to potently neutralized is that long HCDR3 Abs are not favored to be made by the human immune system. In mice the average HCDR3 is 11, and the 460 and 235 abs in this study are HCDR3s of 14 and 15 respectively. Moreover, their epitopes do not match exactly those of the prototype Abs 2F5, 10E8 or DH511. The lack of utility of mice as a model of being able to generate long HCDR3 abs should be mentioned in the discussion.

2. Do Ab 235 and ab 460 bind to lipids?

3. Two suggestions for citations. first Nicely et al. NSMB 17: 1492, 2010, described the structure of the non-neut proximal MPER ab 13H11 that is similar to Ab460 but not exactly. It and most other MPER abs described bind as well to the post fusion 6 helix bundle. Do Abs 235 and 460 bind to gp41 in the post fusion MPER conformation or to the recombinant gp41 6 helix bundle?

Second for IgG3 in MPER may want to also cite with refs 17,18 also ref 13 in line 236 since in 13, all members of the DH511 clone were IgG3.

Reviewer #1 (Remarks to the Author):

I'm mostly satisfied that my comments were addressed, and the authors did some extra experimental work to address my concerns about the glycan shield as well as making the source code used for the model work available to the public on GitHub.

I don't think I have any additional requests for changes, but I'll just note that:

1) It may be sensible for one of the other two reviewers to double check the rebuttal to point #2 (the long response about B cells, lymph nodes, and spleen, is a bit outside my remit).

2) Use caution in putting too much value in the geometric analyses in Figure 6, given lack of physics-based simulations. But I'm happy with the additional effort added in there, plus I think this journal will make the back and forth discussion available for public inspection, so folks can decide on their own.

Tyler Reddy, Los Alamos National Laboratory

Reviewer #2 (Remarks to the Author):

I appreciate the in depth response to mine and the other reviewers comments. Although I am not able to comment on the quality of the experiments/data, the structural work appears important and the results interesting. Unfortunately, the authors basically declined to modify/enhance the modeling despite comments from myself and reviewer 1 that the model is relatively simple as to not very convincingly describe the phenomenon in detail. As is, we do not gain much quantitative insight other than longer is better for crosslinking. I also do not think the authors took very seriously the comments that the paper is hard to read for non-structural biologists familiar with HIV vaccines.

We understand the reviewer's concern regarding quantitative insights. In fact, we took his/her suggestions seriously. In response, the geometric analysis now including Env copy number and Fab geometries suggests how far IgG1 and IgG3 can crosslink spikes at given spike densities. Consequently, we respectively suggest that our current analysis provides some quantitative insights when comparing HIV-1 vs SIV models. The Fig. 6 modeling was driven by experimental results based on differences in breadth and potency of anti-viral activity between IgG1 vs IgG3 bnAbs, searching for the explanation for our experimental data. The modeling is sufficiently unambiguous to offer a cogent explanation for the functional difference between IgG subtypes observed experimentally. While future studies can be performed, given the extensive structural and functional data herein, we believe the current model is appropriate. We thank the reviewer for the model improvement that is now incorporated.

Minor, but there are still several problematic statements in the introduction:

"prevent virus entry into the host" -- is this really what bnAbs do? don't they neutralize/clear virus after it enters the mucosal tissue? SHIV experiments by Liu et al. and Hessel et al. indicate systemic virus spread before clearance.

We modified the sentence from "prevent virus entry into host" to "prevent viral replication" on page 3, line 56.

"HIV-1-positive individuals" - I believe people living with HIV is favored

Based on the reviewer's suggestion, we changed HIV-1 positive individuals to people living with HIV-1 on page 3, line 68.

"outstanding neutralization breadth" - would be important to provide some quantitative numbers here, e.g. in the Huang paper you cite they say 10E8 neutralized 98% of tested viruses, but there must be more up to date information on this

We added a statement on the neutralization breadth of 10E8 and DH511 as follows : The 10E8 and the most potent DH511 clonal lineage bnAb (DH511.2) neutralized 203 of 208 and 206 of 208 viruses (98% and 99%), respectively, in a panel of geographically and genetically diverse HIV-1 Env pseudoviruses with neutralization potency of median IC₅₀ at 0.4 µg/ml for 10E8 and 1.0 µg/ml for DH511.2 on page 3, line. 72-75.

If the authors goal is to influence vaccine research broadly, I do think it would be really important to try to simplify the abstract. The sentences at the end of the introduction summarizing the study are clearer to me than the current abstract. I also attempted to write a version that the authors could borrow from, with the hope it helps:

We thank the reviewer very much for his/her the effort to make our manuscript much approachable to the general readership. We have amalgamated the reviewer's suggestions with our own further modifications in red for the revised abstract.

Broadly neutralizing antibodies (bnAbs) against HIV-1 target conserved **envelope (Env)** epitopes to block viral replication. Here, using structural analyses, we explain why a vaccine targeting the membrane proximal external region (MPER) of HIV-1 **elicits** antibodies with human bnAb-like paratopes **paradoxically** unable to bind HIV-1. Unlike in natural infection, vaccination **with MPER/liposomes** lacks a **necessary** structure-based **constraint** to select for **antibodies with an adequate approach angle**. **Consequently**, the resulting Abs cannot physically access the MPER crawlspace on the virion surface. By studying naturally **arising** Abs, we further **reveal that** flexibility of the human IgG3 hinge **mitigates** the epitope inaccessibility and additionally facilitates **Env** spike protein crosslinking. Our results suggest **that generation of** IgG3 subtype class-switched B cells **is a strategy for anti-MPER** bnAb induction. Moreover, **the findings** illustrate **the need to incorporate topological features of the target epitope in immunogen design**.

Reviewer #3 (Remarks to the Author):

The authors have adequately addressed all outstanding concerns raised during review so the improved/revised manuscript should be accepted for publication.

Reviewer #4 (Remarks to the Author):

The authors have adequately responded to the first three reviewers.

I have only minor comments.

1. the usual reason for lack of MPER antibodies to potently neutralized is that long HCDR3 Abs are not favored to be made by the human immune system. In mice the average HCDR3 is 11, and the 460 and 235 abs in this study are HCDR3s of 14 and 15 respectively. Moreover, their epitopes do not match exactly those of the prototype Abs 2F5, 10E8 or DH511. The lack of utility of mice as a model of being able to generate long HCDR3 abs should be mentioned in the discussion.

We appreciate the reviewer's comment. We agree that eliciting Abs with a long CDRH3 by vaccination would be an important factor for inducing bnAbs. This aspect of CDRH3 length in eliciting

bnAbs is now commented on in the discussion, as indicated in red on page 13, line 401-403.

2. Do Ab 235 and ab 460 bind to lipids?

In response to the reviewer's question, we observed no lipid binding (DOPC/DOPG liposome) by 460 based on ELISA assay, while 235 showed very weak lipid binding with OD value of 0.119 at 1ug/ml concentration. These finding should be compared to 0.8 and 2.0 at 1ug/ml for 2F5 and 4E10, respectively.

3. Two suggestions for citations. first Nicely et al. NSMB 17: 1492, 2010, described the structure of the non-neut proximal MPER ab 13H11 that is similar to Ab460 but not exactly. It and most other MPER abs described bind as well to the post fusion 6 helix bundle. Do Abs 235 and 460 bind to gp41 in the post fusion MPER conformation or to the recombinant gp41 6 helix bundle?

We have modeled the potential binding of 460 and 235 to the post-fusion gp41 state using a structure that contains the fusion peptide, MPER and transmembrane domain (Caillat et al. eLife, 2021, PMID: 33871352). Since the epitope conformation of 460 as well as its accessibility have changed significantly compared to that in its pre-fusion state, 460 cannot bind an individual MPER without major clashes with an adjacent gp41 protomer. This situation is similar to that of 2F5, which is also unable to bind the MPER in the post-fusion state. Likewise, accessibility of 235 to the MPER-C terminal region in the 6-helix bundle structure is not allowed. Based on these analyses, we conclude that it is unlikely for 460 and 235 to bind the MPER in the post-fusion 6-helix bundle structure of gp41.

13H11 antibody is cited on page 8, line 217-221

Second for IgG3 in MPER may want to also cite with refs 17,18 also ref 13 in line 236 since in 13, all members of the DH511 clone were IgG3.

Reference 13 has been added along with ref. 17 and 18 on page 8, line 243.